# Multi-Agent Learning with Heterogeneous Linear Contextual Bandits

**Anh Do**
Johns Hopkins University
`ado8@jhu.edu`

**Thanh Nguyen-Tang**
Johns Hopkins University
`nguyent@cs.jhu.edu`

**Raman Arora**
Johns Hopkins University
`arora@cs.jhu.edu`

## Abstract

As trained intelligent systems become increasingly pervasive, multi-agent learning has emerged as a popular framework for studying complex interactions between autonomous agents. Yet, a formal understanding of how and when learners in heterogeneous environments benefit from sharing their respective experiences is still in its infancy. In this paper, we seek answers to these questions in the context of linear contextual bandits. We present a novel distributed learning algorithm based on the upper confidence bound (UCB) algorithm, which we refer to as H-LINUCB, wherein agents cooperatively minimize the group regret under the coordination of a central server. In the setting where the level of heterogeneity or dissimilarity across the environments is known to the agents, we show that H-LINUCB is provably optimal in regimes where the tasks are highly similar or highly dissimilar.

## 1 Introduction

Heterogeneous multi-agent systems enable agents to work together and coordinate their actions to solve complex problems. These systems are inherently scalable, as they can distribute the computational load across multiple agents. This scalability allows the system to handle large and sophisticated tasks beyond the capabilities of a single agent. Despite the potential of multi-agent systems, it poses the following fundamental challenges.

- **Statistical challenge.** Each agent's reward distribution may vary, meaning that different agents receive different rewards for the same action. This heterogeneity in reward distributions introduces complexity and makes coordination among agents more difficult. Furthermore, Wang et al. [2021] point out that an ineffective use of shared data could lead to a significant negative impact on the overall performance. In particular, sharing experiences amongst agents may hinder the system's performance if the tasks are too dissimilar [Rosenstein, 2005, Brunskill and Li, 2013].

- **Computational complexity.** Coordinating the decisions of numerous agents and performing complex computations pose challenges in terms of computational resources, time constraints, and algorithmic scalability.

- **Communication cost.** Efficient communication is also a fundamental challenge for a large-scale multi-agent system. As the number of agents in a system increases, the complexity of interactions between agents grows exponentially. Managing the interaction of numerous agents and making decisions in a timely manner becomes increasingly difficult.

Several works address these challenges for heterogeneous multi-agent systems, including federated linear bandits [Huang et al., 2021, Li and Wang, 2022], clustering bandits [Gentile et al., 2014, Ghosh et al., 2023], multi-task linear bandits [Hu et al., 2021, Yang et al., 2021]. However, these works either rely on some special structure of the parameter [Li and Wang, 2022], e.g., a low-rank structure [Hu

et al., 2021] or make different assumptions, e.g., stochastic contexts, finite decision set [Huang et al., 2021, Ghosh et al., 2023], etc.

In this work, we provide a general notion of heterogeneous multi-agent linear contextual bandits ($\varepsilon$-MALCB) and give analytical results under different regimes of heterogeneity. Specifically, we study a model that consists of $M$ agents. Each agent $i \in [M]$ plays a $d$-dimensional linear contextual bandit, parametrized by $\theta_i$ (see Section 3 for more details), for $T$ rounds. We capture the heterogeneity by a dissimilarity parameter $\varepsilon > 0$ such that $\|\theta_i - \theta_j\|_2 \le \varepsilon$, for all $i, j \in [M]$. Notably, we do not assume any special structure on the linear parameter; and we allow the decision set to be infinite and possibly chosen adversarially.

**Motivating application.** Consider a personalized recommendation system for online advertisements [Li et al., 2010, Bouneffouf et al., 2020, Ghosh et al., 2023]. Here, the platform needs to be adaptive to user preferences and maximize total user clicks based on user-click feedback. Each ad can be represented as a context vector, encoding information such as publisher, topic, ad content, etc. The inner product of the ad's context vector and user preference represents the alignment. A higher inner product value indicates greater relevance of the ad. Furthermore, the recommended ads must be personalized to accommodate user preference differences. One naive approach would involve solving a separate linear contextual bandit problem for each user. However, we can pose the following question: Can we enhance the system's performance by pooling data from other users? If so, to what extent of user heterogeneity can we achieve that?

**Contributions.** We make the following contributions in this paper.

- First, we formulate the heterogeneous multi-agent linear contextual bandits as $\varepsilon$-MALCB problem, building on the classic notion of heterogeneity in multiarmed bandits (MABs) Wang et al. [2021]. Our notion of heterogeneity is natural and captures many settings in real-world applications.

- Second, when the level of dissimilarity is known, we propose a distributed algorithm, namely H-LINUCB which achieves a regret of $\tilde{\mathcal{O}}(d\sqrt{MT} + \min\{\varepsilon dMT, dM\sqrt{T}\})$ under the coordination of a central server. We discuss in detail how to handle the dissimilarity and introduce a criterion for stopping collaboration when the level of dissimilarity is high. We show that under the regime of low dissimilarity, we can still achieve a regret of $\tilde{\mathcal{O}}(d\sqrt{MT})$, which is the same regret rate as if $M$ agents collaborate to solve the *same* task. In this regime, H-LINUCB outperforms independent learners improving by a factor of $\sqrt{M}$, from $\tilde{\mathcal{O}}(dM\sqrt{T})$ to $\tilde{\mathcal{O}}(d\sqrt{MT})$. This is significant when we have a large number of agents.

- Third, we complement the upper bound with a lower bound of $\Omega(d\sqrt{MT} + \min\{\varepsilon MT, dM\sqrt{T}\})$. This suggests that our theoretical guarantees are tight in settings where tasks are highly similar or highly dissimilar.

- Finally, we validate our theoretical results with numerical simulations on synthetic data. When the level of dissimilarity is small, H-LINUCB outperforms independent learning. When the level of dissimilarity is high, our simulation shows that blindly using shared data can lead to linear regret, emphasizing the importance of the criterion we propose for when to stop the collaboration.

## 2 Related work

The classic linear bandits have a rich literature in both theory and application; see, for example, [Abbasi-Yadkori et al., 2011, Li et al., 2010, Chu et al., 2011, Chatterji et al., 2020, Rusmevichientong and Tsitsiklis, 2010, Bouneffouf et al., 2020, Mahadik et al., 2020], to name a few. With a surge in distributed computing, multi-agent systems have shown their potential and gained more attention in recent years. A large body of works dedicated to studying the homogeneous setting of multiple collaborating agents solve a global linear bandits problem [Wang et al., 2020, Dubey and Pentland, 2020, Moradipari et al., 2022, Mitra et al., 2022, Martínez-Rubio et al., 2019, Chawla et al., 2022].

The problem of multi-agent linear bandits in heterogeneous environments, on the other hand, has received limited attention. Soare et al. [2014] were amongst the first to study heterogeneous linear

bandits; however, the focus in that work is not on group regret, and the authors only consider the setting where tasks are similar. More recently, Huang et al. [2021] proposed an algorithm with a novel multi-agent G-Optimal design. They assume that the heterogeneity comes from the contexts associated with each agent, but agents can still collaborate since they share the same arm parameters. Li and Wang [2022] consider an extension where they assume that each agent's parameter has two components – a shared global component and an individual local component. This formalization requires agents to work on their respective tasks (the local component) but still allows agents to collaborate on the common task (the global component). Wang et al. [2021] study heterogeneity of the Bernoulli MABs problem and provide guarantees for both cases when the level of heterogeneity is known and unknown.

A related line of work studies heterogeneous linear bandits through clustering [Gentile et al., 2014, Li et al., 2016, 2019, Korda et al., 2016, Ghosh et al., 2023]. These works give a guarantee based on the clustering structure of the different linear bandit problems – agents belonging to the same cluster will likely achieve the highest collaboration gain. We do not make any assumption about the "clusterability" of different bandit problems we may encounter. A yet another approach focuses on multi-task linear bandits, wherein we solve multiple different but closely related linear bandits tasks [Yang et al., 2021, 2022, Hu et al., 2021, Cella et al., 2023, Du et al., 2023]. In particular, these works rely on the assumption that all tasks share a common $k$-dimensional representation, where $k \ll d$. Then, pooling data from different bandits helps learn a good representation and reduces the statistical burden of learning by reducing the linear bandit problem in $d$ dimensions to a $k$-dimensional setting. We do not consider multi-task learning here.

Our formulation of heterogeneous contextual linear bandits is similar to that of misspecified and corrupted bandits setting (Remark 3.1 for more details) [Ghosh et al., 2017, Lattimore and Csaba, 2020, Takemura et al., 2021, Foster et al., 2020, He et al., 2022]. It is then natural to ask if we can apply the techniques from that part of the literature to deal with the dissimilarity between different bandits in a heterogeneous setting. However, there is a fundamental difference in how the two problems manifest themselves. While misspecification may be typically unavoidable in many settings, in a heterogeneous bandit setting, an agent can always choose to rely solely on its own data if it finds that the data from other agents are too dissimilar.

## 3 Preliminaries

**Multi-Agent Linear Contextual Bandits.** We consider a multi-agent learning setting with $M$ agents. At each round $t \in [T]$, each agent $m \in [M]$ picks an action (context)[1] $x_{m,t} \in \mathcal{D}_{m,t}$, where $\mathcal{D}_{m,t} \subseteq \mathbb{R}^d$ is a given decision set. The agent $m$ receives reward $y_{m,t} = x_{m,t}^\top \theta_m + \eta_{m,t}$, where $\theta_m \in \mathbb{R}^d$ is an unknown but fixed parameter and $\eta_{m,t}$ is sub-Gaussian noise. Let $\mathcal{F}_t$ denote the filtration, i.e., the $\sigma$-algebra, induced by $\sigma(\{x_{m,k}\}_{m \in [M], k \leq t+1}, \{\eta_{m,k}\}_{m \in [M], k \leq t})$.

**Regret.** Our goal is to design algorithms for multi-agent linear contextual bandits that achieve a small group regret defined as

$$\mathcal{R}(M,T) = \sum_{t=1}^{T} \sum_{m=1}^{M} \left( \max_{x \in \mathcal{D}_{m,t}} \langle x, \theta_m \rangle - \langle x_{m,t}, \theta_m \rangle \right).$$

**Assumption 3.1.** Without loss of generality, we assume that,

1. **Bounded parameters**: $\|\theta_m\|_2 \leq 1, \|x\|_2 \leq 1, \forall x \in \mathcal{D}_{m,t}, m \in [M], t \in [T]$.

2. **Sub-Gaussian noise**: $\eta_{m,t}$ is conditionally zero-mean 1-sub-Gaussian random variable with respect to $\mathcal{F}_{t-1}$.

We note that the assumptions above are standard in linear bandits literature [Abbasi-Yadkori et al., 2011, Hu et al., 2021, Huang et al., 2021]. Further, it is straightforward to let $\|\theta_m\| \leq B$, for some constant $B$, by appropriately scaling the rewards. We make no additional assumptions on the context. The decision set could be infinite, and given to each agent possibly adversarially.

---

[1] Throughout the paper, we use the terms *action* and *context* interchangeably.

**Definition 3.1.** *(ε-MALCB Problem) A multi-agent linear contextual bandits problem is said to be an ε-MALCB problem instance, if for any two agents $i, j \in [M]$, $\|\theta_i - \theta_j\|_2 \leq \varepsilon$, for an $\varepsilon \geq 0$. We call $\varepsilon$ the dissimilarity parameter.*

**Definition 3.2.** *(Homogeneous setting) A multi-agent linear contextual bandits problem is homogeneous, if it is an ε-MALCB with $\varepsilon = 0$, i.e., $\theta_i = \theta_j$, for all $i, j \in [M]$.*

Given the bound on the parameters, we have that $\|\theta_i - \theta_j\|_2 \leq 2$ for any $i, j \in [M]$. Therefore, it suffices to only consider the case where $\varepsilon \in [0, 2]$.

**Remark 3.1.** (Misspecified structure) Under the Assumption 3.1, for any two agents $i, j$ we have that $\theta_i^\top x - \varepsilon \leq \theta_j^\top x \leq \theta_i^\top x + \varepsilon$. Then, $\mathbb{E}[y_{j,x}] = \theta_i^\top x + \Delta(x)$, for $\Delta(x) \in [-\varepsilon, \varepsilon]$. This represents a misspecified structure wherein agent $i$ receives the reward $y_{j,x}$ from agent $j \neq i$.

**Remark 3.2.** (Recover the $\varepsilon$-MPMAB of Wang et al. [2021]). We note that the $\varepsilon$-MPMAB is a special case of $\varepsilon$-MALCB. Define the mean reward of $K$ arms for agent $m$ as $\theta_m = [\mu_1^m, \ldots, \mu_K^m]$. Then, the reward for arm $k$ at round $t$ is $y_{t,k}^m = \theta_i^\top e_k + \eta_t$. The decision set $\mathcal{D} = \{e_1, \ldots, e_K\}$ are the standard basis vectors. This is a fixed set of arms, given to all agents at each round. The dissimilarity parameter $\varepsilon$ is defined as: $\|\theta_i - \theta_j\|_\infty \leq \varepsilon$ for all $i, j \in [M]$.

Nonetheless, the results in Wang et al. [2021] are not directly comparable to ours since the dissimilarity parameter $\varepsilon$ hides inside the size of the set of *subpar* arms $|\mathcal{I}_\varepsilon|$.[2] Furthermore, Wang et al. [2021] give guarantees in a full-communication setting, in which each agent has full access to the past data of all other agents at every round.

**Remark 3.3.** There are other formulations that also capture the heterogeneity in multi-agent linear bandits. Huang et al. [2021] consider a multi-agent linear bandits setting with a fixed size decision set, containing $K$ actions, $\{\theta_a\}_{a=1}^K$, which is unknown to the agents. Each agent $i$ is associated with $K$ different contexts $\{x_{i,a}\}_{a=1}^K$. At reach round, each agent $i$ picks an action $a \in [K]$, and receives reward $r_{i,a} = x_{i,a}^\top \theta_a + \eta_{i,a}$. Since $x_{i,a}$ can vary for different agents, this captures the heterogeneity across agents. It also allows for collaboration across agents since they share the same decision set.

Li and Wang [2022] assume that each agent parameter $\theta$ has a special structure that consists of a shared global component and a unique local component. The reward of agent $i$ can be given as, $r_{i,x} = \begin{bmatrix} \theta^{(g)} \\ \theta^{(i)} \end{bmatrix}^\top \begin{bmatrix} \mathbf{x}^{(g)} \\ \mathbf{x}^{(l)} \end{bmatrix}$. We do not assume such special structure of the linear parameter.

The notion of $\varepsilon$-MALCB is in the worst-case sense. One can imagine an $\varepsilon$-MALCB instance of $M$ agents, such that $M - 1$ agents have identical linear parameter, i.e. $\theta_i = \theta_j, \forall i, j \in [M - 1]$, and the parameter of the last agent $\|\theta_M - \theta_{M-1}\|_2 = \varepsilon$. With this type of instance, for a large $M$, we can simply use DISLINUCB of Wang et al. [2021] and achieve nearly optimal regret. Clustering bandits framework could also be used for this $\varepsilon$-MALCB instance since it presents a strong cluster structure. Even though our formulation takes a pessimistic approach but it can handle the case that bandits are largely unclustered.

Our goal is to design a system such that its performance is no worse than running $M$ independent bandit algorithms for each user (zero collaboration). The system should also be adaptive to the heterogeneity in the problem instance, i.e., it should automatically leverage any structure in the problem parameters $\{\theta_m\}_{m=1}^M$ to collaboratively solve all bandit problems at a "faster" rate. To benchmark the performance of such a system, we consider the following baseline.

**Independent Learners (IND-OFUL).**    We establish a baseline algorithm in which each agent independently runs an optimal linear contextual bandits algorithm (OFUL, [Abbasi-Yadkori et al., 2011]) without any communication. Each agent incurs $\tilde{\mathcal{O}}(d\sqrt{T})$ regret, and $\tilde{\mathcal{O}}(dM\sqrt{T})$ group regret.

**Notation.**    We denote the weighted norm of vector $x$ w.r.t. matrix $A$ (Mahalanobis norm) as $\|x\|_A = \sqrt{x^\top A x}$. We write $A \succeq B$ iff $A - B$ is a positive semi-definite matrix. We use $\tilde{\mathcal{O}}(\cdot)$ to hide the polylogarithmic factors in standard Big O notation.

---

[2]ROBUSTAGG($\varepsilon$) algorithm achieves $\tilde{\mathcal{O}}(\sqrt{\mathcal{I}_\varepsilon MT} + M\sqrt{(|\mathcal{I}_\varepsilon| - 1)T} + M|\mathcal{I}_\varepsilon|)$ regret when $\varepsilon$ is known. The subpar arms $\mathcal{I}_\varepsilon$ is defined in Wang et al. [2021, Section 3.2].

# 4 Main Results

In this section, we present H-LINUCB, a UCB-style algorithm for $\varepsilon$-MALCB problem, and give guarantees for the case when the dissimilarity $\varepsilon$ is known to the agents. In Section 4.2, we present a lower bound and discuss the implication of our results in different regimes of dissimilarity. We defer the detailed proof to the Appendix.

---

**Algorithm 1** H-LINUCB

---

**Input:** Dissimilarity parameter $\varepsilon$, number of agents $M$, number of rounds $T$, regularization parameter $\lambda$, dimension $d$, confidence parameters $\beta_t, \forall t \in [T]$, weight threshold parameter $\alpha$, collaboration budget $\tau$, synchronization threshold $D$

1: $t_{syn} \leftarrow 1$                            $\triangleright$ $t_{syn}$ is the index of the last synchronized round
2: $V_{syn} \leftarrow 0, b_{syn} \leftarrow 0$ $\triangleright$ $V_{syn}, b_{syn}$ store the relevant statistics of all agents after synchronization
3: $V_{epoch,m} \leftarrow 0, b_{epoch,m} \leftarrow 0, \forall m \in [M]$     $\triangleright$ $V_{epoch,m}, b_{epoch,m}$ store the relevant statistics of agent $m$ in the current epoch
4: **for** $t = 1, \cdots, T$ **do**
5:      **for** Agent $m = 1, \cdots, M$ **do**
6:          **if** $t = \tau$ **then**
7:              $V_{syn} \leftarrow 0, b_{syn} \leftarrow 0$
8:              $V_{epoch,m} \leftarrow 0, b_{epoch,m} \leftarrow 0$
9:          **end if**
10:         $V_{m,t} \leftarrow \lambda I + V_{syn} + V_{epoch,m}$
11:         $\hat{\theta}_{m,t} \leftarrow V_{m,t}^{-1} (b_{syn} + b_{epoch,m})$
12:         Construct the confidence ellipsoid $\mathcal{C}_{m,t} = \left\{ \theta \in \mathbb{R}^d : \left\| \hat{\theta}_{m,t} - \theta \right\|_{V_{m,t}} \leq \beta_t \right\}$
13:         $(x_{m,t}, \tilde{\theta}_{m,t}) = \arg\max_{(x,\theta) \in \mathcal{D}_{m,t} \times \mathcal{C}_{m,t}} \langle \theta, x \rangle$
14:         Play $x_{m,t}$ and get reward $y_{m,t}$
15:         $w_{m,t} \leftarrow \mathbb{1}\left[ t < \tau \right] \min \left( 1, \alpha / \|x_{m,t}\|_{V_{m,t}^{-1}} \right) + \mathbb{1}[t \geq \tau]$
16:         $V_{epoch,m} \leftarrow V_{epoch,m} + w_{m,t} x_{m,t} x_{m,t}^\top$
17:         $b_{epoch,m} \leftarrow b_{epoch,m} + w_{m,t} x_{m,t} y_{m,t}$
18:         **if** $\log \left[ \det(V_{m,t} + w_{m,t} x_{m,t} x_{m,t}^\top) / \det(\lambda I + V_{syn}) \right] \cdot (t - t_{syn}) \geq D$ and $t < \tau$ **then**
19:             Send a synchronization signal to the server to start a communication round
20:         **end if**
21:         **if** A communication round is started **then**
22:             Agent $i$ sends $V_{epoch,i}, b_{epoch,i}$ to the server, $\forall i \in [M]$
23:             Server computes $V_{syn} \leftarrow V_{syn} + V_{epoch,i}, b_{syn} \leftarrow b_{syn} + b_{epoch,i}, \forall i \in [M]$
24:             Server sends $V_{syn}, b_{syn}$ back to all agents
25:             $V_{epoch,i} \leftarrow 0; b_{epoch,i} \leftarrow 0; \forall i \in [M]$      $\triangleright$ Reset $V_{epoch,i}, b_{epoch,i}$ for the new epoch
26:             $t_{syn} \leftarrow t$
27:         **end if**
28:      **end for**
29: **end for**

---

## 4.1 H-LINUCB Algorithm

H-LINUCB is a distributed UCB-style algorithm (see Algorithm 1 for pseudocode), in which agents work cooperatively under the coordination of a central server.

H-LINUCB has two learning phases: the *collaboration* phase (for rounds $t \in \{1, \ldots, \tau - 1\}$) and the *independent learning* phase (for rounds $t \in \{\tau, \ldots, T\}$), where $\tau \leq T$ is the collaboration budget. Intuitively, our two-phase learning framework ensures that the agents stop collaboration after $\tau$ rounds lest they incur a linear regret in bandit environments with large dissimilarity. Naturally, then, the parameter $\tau$ should depend on the dissimilarity parameter, $\varepsilon$. We give an optimal choice of $\tau$ in Theorem 4.1.

At each round $t < \tau$ (the collaboration phase), each agent's data is weighted to adapt to the dissimilarity across different agents (Line 15). Then, each agent uses the weighted data to construct

its Confidence Ellipsoid (Line 12) and makes a decision following the optimism principle (Line 13). When a certain condition is met (Line 18), data is pooled and synchronized across the agents. Starting from round $\tau$, all collaboration ceases and each agent enters the independent learning mode and runs an independent copy of the OFUL algorithm [Abbasi-Yadkori et al., 2011] locally for the last $T - \tau + 1$ rounds.

We note that H-LINUCB builds upon DISLINUCB of Wang et al. [2020, Protocol 8] with the following modifications:

- We scale each agent's data using the weight $\min(1, \alpha/\|x_{m,t}\|_{V_{m,t}^{-1}})$, which we adopt from He et al. [2022], to handle the dissimilarity across different agents (Line 15).
- We only allow collaboration until round $\tau$ (Line 18). The value of $\tau$ depends on the dissimilarity parameter, which we assume is given.
- We reset the variables $V_{syn}, b_{syn}, V_{epoch,m}, b_{epoch,m}$ at round $\tau$ (Lines 6-9), where each agent switches to the independent learning mode. Here, *epoch* refers to the time period between two consecutive synchronization rounds.

Each agent uses all of the data available to them at each round to construct the *Confidence Ellipsoid* $\mathcal{C}_{m,t}$ using the result in Lemma 4.1. Given the confidence ellipsoid, the agent chooses the action optimistically: $(x_{m,t}, \tilde{\theta}_{m,t}) = \arg\max_{(x,\theta) \in \mathcal{D}_{m,t} \times \mathcal{C}_{m,t}} \langle \theta, x \rangle$. During the collaboration phase, if the variation in the volume of the ellipsoid exceeds a certain synchronization threshold, $D$, it triggers a synchronization condition (Lines 18-20). Subsequently, the central server commences the synchronization procedure to update $V_{syn}, b_{syn}$ across all participating agents (Lines 21-27. The optimal value of $D$ depends on the number of agents $M$, dimension $d$, and the collaboration budget $\tau$.

The weight $\min(1, \alpha/\|x_{m,t}\|_{V_{m,t}^{-1}})$ is a truncation of the inverse bonus, where $\alpha > 0$ is a threshold parameter that shall be optimized later. When $x_{m,t}$ is not explored much, we have a large exploration bonus $\|x_{m,t}\|_{V_{m,t}^{-1}}$ (low confidence). Hence, the algorithm will put a small weight on it to avoid a large regret due to stochastic noise and misspecification. When $\|x_{m,t}\|_{V_{m,t}^{-1}}$ is small (high confidence), H-LINUCB puts a large weight on it, and it can be as large as one [He et al., 2022].[3] We note that using this weighting could have a significant negative impact on the performance if we are not careful. Several recent studies show that we can incur a regret of $\tilde{\mathcal{O}}(d\sqrt{T} + \varepsilon\sqrt{d}T)$ and $\tilde{\mathcal{O}}(d\sqrt{T} + \varepsilon dT)$ for misspecified and corrupted linear bandits, respectively [Lattimore and Csaba, 2020, Takemura et al., 2021, Foster et al., 2020, He et al., 2022].[4] However, a direct application of an algorithm designed for misspecified/corrupted linear bandits to our setting can lead to linear regret when $\varepsilon = \Theta(1)$. This is significantly worse as compared to naive independent learning, which always achieves sub-linear $\tilde{\mathcal{O}}(dM\sqrt{T})$ regret.

We emphasize that the condition $(t < \tau)$ in Line 18 is crucial to avoid linear regret for H-LINUCB in the regime of large $\varepsilon$. For example, for $\varepsilon = \Theta(1)$, $\tau = T$, Algorithm 1 incurs $\tilde{\mathcal{O}}(dMT)$ regret, which is linear in term of $MT$. Furthermore, Theorem 4.3 indicates that there exists an instance of $\varepsilon$-MALCB such that any algorithm incurs at least $\Omega(dM\sqrt{T})$ regret. Then, each agent playing OFUL independently would be enough to achieve an optimal $\tilde{\mathcal{O}}(dM\sqrt{T})$ regret. This suggests that we get a tighter upper bound if we cease collaboration; we discuss the stopping criterion and the choice of $\tau$ in Theorem 4.2.

**Communication protocol.** We use a star-shaped communication network where $M$ agents can interact with a central server [Wang et al., 2020, Dubey and Pentland, 2020]. Each agent communicates with the server by uploading and downloading its data but does not communicate directly with each other. The communication will be triggered only if any agent has enough new data since the last synchronization round. Finally, we assume no latency, or error in the communication between the central server and agents.

---

[3]The technique of using the exploration bonus to control misspecification is also used in Zhang et al. [2023].

[4]Takemura et al. [2021], Foster et al. [2020] give guarantees for when the misspecification level is unknown. The CW-OFUL algorithm of He et al. [2022] has $\tilde{\mathcal{O}}(d\sqrt{T} + dC)$ regret, where $C$ is the total amount of corruption. Setting $C = \varepsilon T$, where $\varepsilon$ is the level of misspecification at each round, gives us $\tilde{\mathcal{O}}(d\sqrt{T} + \varepsilon dT)$ regret.

**Remark 4.1.** Wang et al. [2020] show that DISLINUCB can rely on old data and produce nearly optimal policy without much communication, only incurring logarithmic factors in the final regret. H-LINUCB has the same communication cost of $\mathcal{O}(M^{1.5}d^3)$, as DISLINUCB, which does not depend on the horizon $T$.

When an agent uses data from other agents, due to the dissimilarity, we need to adjust our confidence bound to make sure the true linear parameter $\theta_m$ lies in the defined ellipsoid with high probability. The next result shows how to construct such a *Confidence Ellipsoid*.

**Lemma 4.1.** *(Confidence Ellipsoid). With probability at least* $1 - M\delta_1 - M\delta_2$*, for each agent* $m \in [M]$*,* $\theta_m$ *lies in the confidence set,*

$$\mathcal{C}_{m,t} = \left\{ \theta \in \mathbb{R}^d : \left\| \hat{\theta}_{m,t} - \theta \right\|_{V_{m,t}} \le \beta_t \right\},$$

*where*

$$\beta_t = \begin{cases} \sqrt{\lambda} + \alpha \varepsilon M t + \sqrt{d \log\left(\frac{1 + Mt/(\lambda d)}{\delta_1}\right)}, \text{ for } t < \tau, \\ \sqrt{\lambda} + \sqrt{d \log\left(\frac{1 + t/(\lambda d)}{\delta_2}\right)}, \text{ for } t \ge \tau. \end{cases}$$

Note that the result above provides two separate confidence bounds for $\theta_m$, one for the period before round $\tau$ and the other one for after round $\tau$. Before round $\tau$, agent $m$ will use all the data from other agents to construct $\mathcal{C}_{m,t}$. The proof follows by first bounding $\left\| \hat{\theta}_{m,t} - \theta_m \right\|_{V_{m,t}(\lambda)}$ as

$$\left\| \hat{\theta}_{m,t} - \theta_m \right\|_{V_{m,t}(\lambda)} \le \underbrace{I_1}_{\text{Regularization term}} + \underbrace{\left\| \sum_{i \ne m} \sum_{k=1}^{t_s} w_{i,k} x_{i,k} x_{i,k}^\top (\theta_i - \theta_m) \right\|_{V_{m,t}^{-1}(\lambda)}}_{I_2:\text{Dissimilarity term}} + \underbrace{I_3}_{\text{Noise term}}.$$

Here, $I_1$ is a bounded regularization term, $I_3$ is a noise term that can be bounded by self-normalization lemma (Lemma C.4). Finally, the dissimilarity term $I_2$ is bounded from above by $\alpha \varepsilon M t$ using the definition of dissimilarity $\|\theta_i - \theta_m\| \le \varepsilon$, and applying a similar argument as He et al. [2022] and the choice of the weight $w_{m,t} = \min(1, \alpha/\|x_{m,t}\|_{V_{m,t}^{-1}})$. For the phase after round $\tau$, we use the same argument as Abbasi-Yadkori et al. [2011] to construct the confidence bound.

Next, we present the group regret upper bound of Algorithm 1 up to round $\tau$.

**Theorem 4.1.** *Given Assumption 3.1, $T \ge 1$, any $\tau \le T$, and $\delta_1 > 0$, setting $\lambda = 1$ and $\alpha = \frac{\sqrt{d}}{\varepsilon M \tau}$ in the upper bound of $\beta_t$ on the confidence interval according to Lemma 4.1 $\forall t \in [T]$, and setting the synchronization threshold $D = \tau \log(M\tau)/(dM)$, we have that with probability at least $1 - M\delta_1$, the group regret of Algorithm 1 up to round $\tau$, is bounded as*

$$\mathcal{R}(M, \tau) \le 20\sqrt{2} \left( d\sqrt{M\tau} \xi_\tau^2 + \varepsilon dM\tau \xi_\tau^{1.5} \right),$$

*where $\xi_t = \log\left(\frac{1 + Mt/(\lambda d)}{\delta_1}\right)$.*

Theorem 4.1 shows that Algorithm 1 incurs $\tilde{\mathcal{O}}(d\sqrt{M\tau})$ regret in the first term (which is the same order as a single agent playing for $M\tau$ rounds) plus a penalty of using the data from other agents in the order of $\tilde{\mathcal{O}}(\varepsilon dM\tau)$. The regret of $\tilde{\mathcal{O}}(d\sqrt{M\tau})$ is unavoidable for any regime of $\varepsilon$, and this rate is known to be optimal in the case of homogeneous multi-agent. Since we use the technique from He et al. [2022] to handle the dissimilarity, the corruption amount of each round is $\varepsilon$, and a total corruption of $\varepsilon dM\tau$ when the central server allows to collaborate up to round $\tau$. To the best of our knowledge, we are not aware of any UCB-based algorithm for misspecified linear bandits in the setting of infinite arms. We expect that employing a misspecified linear bandits algorithm would achieve a regret of $\tilde{\mathcal{O}}(d\sqrt{MT} + \varepsilon\sqrt{d}MT)$, which is tighter by a factor of $\sqrt{d}$. It is worth noting that the CW-OFUL algorithm in He et al. [2022] is designed for handling corruption, whether it can achieve $\tilde{\mathcal{O}}(d\sqrt{T} + \varepsilon\sqrt{d}T)$ in a misspecified setting remains an open question.

We now present our main result giving an upper bound on the group regret of H-LINUCB.

**Theorem 4.2.** *Given Assumption 3.1, $T \geq 1$, let $\lambda = 1, \alpha = \frac{\sqrt{d}}{\varepsilon M \tau}, \delta_1 = \delta_2 = \frac{1}{M^2 T}$ in the upper bound of $\beta_t$ on the confidence interval (see Lemma 4.1) $\forall t \in [T]$. Let $\tau = \min(\lfloor \frac{1}{2\varepsilon^2} \rfloor, T)$, and let the synchronization threshold $D = \tau \log(M\tau)/(dM)$. Then, the expected group regret of Algorithm 1 is bounded as*

$$\mathbb{E}[\mathcal{R}(M,T)] \leq 320\sqrt{2}\left(d\sqrt{MT} + 2\min\left\{\varepsilon dMT, dM\sqrt{T}\right\}\right)\log^2(MT).$$

Here, $\tau$ is the maximum round that the central server allows communication. After that, all agents switch to independent learning. By choosing $\tau = \min(\lfloor \frac{1}{2\varepsilon^2} \rfloor, T)$, agents fully cooperate in the regime $\varepsilon \in [0, \frac{1}{\sqrt{2T}}]$ if $T \leq \frac{1}{2\varepsilon^2}$, and gradually reduce $\tau$ as $\varepsilon$ increases from $\frac{1}{\sqrt{2T}}$ to $+\infty$. This is important for avoiding a linear regret since when $\varepsilon$ is large, most of the regret comes from the $\varepsilon dMT$ term and dominates the $d\sqrt{MT}$ term. The condition on Line 6 also discards all of the synchronized data. In the extreme case, when $\varepsilon > 1$, there is no collaboration happening due to the condition in Line 18 failing at every round. In other words, H-LINUCB behaves like IND-OFUL.

In the other extreme case, when $\varepsilon = 0$, all agents solve identical linear bandits. The weight in Line 15 always evaluates to its minimum value of 1 for all $t \in [T]$ since $\alpha = \frac{\sqrt{d}}{\varepsilon MT} \to +\infty$. We have $t < \varepsilon^2$ for all rounds. Therefore, the reset condition in Line 6 is never triggered and H-LINUCB behaves exactly like DISLINUCB, achieving a regret of $\tilde{\mathcal{O}}(d\sqrt{MT})$, which is optimal up to some logarithmic factors.

Theorem 4.2 suggests that the upper bound of H-LINUCB is tighter than IND-OFUL for all $\varepsilon$.

**Remark 4.2.** Ghosh et al. [2023] also propose a personalized algorithm (PMLB) for the heterogeneous multi-agent linear bandits; however, our problem setting is fundamentally different than that of Ghosh et al. [2023]. They consider a finite action set and impose a strong distributional assumption on how contexts are generated, i.e., the stochastic context $x_{i,t}$ for each action $i$ and at each round $t$ is zero-mean and forms a positive-definite covariance matrix. In stark contrast, we consider the adversarial setting where the context set is *adversarially* generated at each round (and thus, the associated action set can be infinite and arbitrary). This renders the algorithm and guarantees of Ghosh et al. [2023] inapplicable in our setting and, thus, requires a completely different treatment. Those assumptions of Ghosh et al. [2023] are crucial for them to obtain the $\tilde{\mathcal{O}}(T^{1/4})$ bound (for $\varepsilon = 0$). We note that this bound is not information-theoretically possible in our adversarial setting; the minimax lower bound in such settings is $\Omega(\sqrt{T})$ (Theorem 4.3).

## 4.2 Lower bound

In this section, we present a lower bound result for the $\varepsilon$-MALCB problem. We denote $\mathcal{R}_{\mathcal{A},\mathcal{I}}(M,T)$ as a regret of algorithm $\mathcal{A}$ on a problem instance $\mathcal{I}$ of $M$ agents run for $T$ rounds.

**Theorem 4.3.** *Let $\mathcal{I}(\varepsilon)$ denote the class of $\varepsilon$-MALCB problem instances that satisfy the Assumption 3.1. Then for any $d, M, T \in \mathbb{Z}^+$ with $\frac{d}{2} \leq T, \frac{d^2}{48} \leq T, \forall \varepsilon \geq 0$, we have the following,*

$$\inf_{\mathcal{A}} \sup_{\mathcal{I} \in \mathcal{I}(\varepsilon)} \mathcal{R}_{\mathcal{A},\mathcal{I}}(M,T) = \Omega\left(d\sqrt{MT} + \min\left\{\varepsilon MT, dM\sqrt{T}\right\}\right).$$

The first term is a straightforward observation that solving an $\varepsilon$-MALCB is at least as hard as solving a single linear bandits for $MT$ rounds, or $M$ agents solving identical bandits for $T$ rounds. The second term suggests that we pay an additional regret of $\varepsilon MT$ for a *small* $\varepsilon \in [0, \frac{d}{\sqrt{T}}]$, and $\Omega(dM\sqrt{T})$ for a *large* $\varepsilon \geq \frac{d}{\sqrt{T}}$. We note that $\Omega(dM\sqrt{T})$ is also the lower bound of IND-OFUL when each agent incurs a regret of at least $\Omega(d\sqrt{T})$. We believe that the analysis of the lower bound could be tightened by using the arguments from misspecified bandits literature, achieving a lower bound of $\Omega(d\sqrt{MT} + \min\{\varepsilon\sqrt{d}MT, dM\sqrt{T}\})$.

The lower bound suggests that our upper bound is tight up to logarithmic factors in the following extreme regimes, (i) $\varepsilon \in [0, \frac{1}{\sqrt{MT}}]$, where $\mathcal{R}(M,T) = \Theta(d\sqrt{MT})$; (ii) $\varepsilon \in [\frac{d}{\sqrt{T}}, +\infty]$, where $\mathcal{R}(M,T) = \Theta(dM\sqrt{T})$. In regime (i), all agents solve tasks that are similar to one another, yielding the highest collaborative gain. In regime (ii), tasks are highly dissimilar, H-LINUCB turns off the collaboration and lets agents solve their own tasks individually.

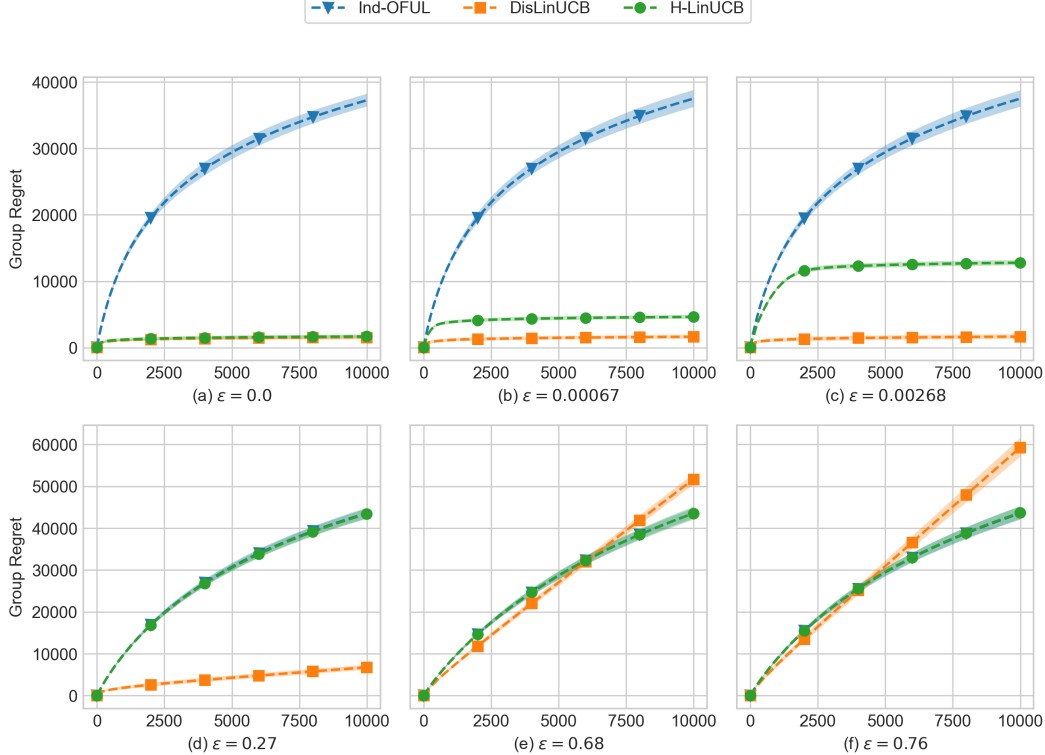

Figure 1: Simulation on synthetic data with $M = 60, d = 30, T = 10000$.

Finally, in the regime that $\varepsilon \in (\frac{1}{\sqrt{MT}}, \frac{d}{\sqrt{T}})$, our results illustrate the interpolation between two extremes. In this regime, our upper bound presents a gap of $d$ in the dissimilarity term $\varepsilon dMT$ compared to $\varepsilon MT$ in the lower bound.

The key idea in the proof of Theorem 4.3 is based on an information-theoretic lower bound of Lemma C.1, wherein we extend the result from (single-agent) linear bandits to *heterogeneous* multi-agent linear bandits. Here, we give the lower bound result without any constraints on the communication, hence, this is also the lower bound of the H-LINUCB algorithm.

## 5   Numerical Simulations

In this section, we provide some numerical simulations to support our theory. Our goal is to address the following question: how does H-LINUCB perform in three different regimes of dissimilarity: (i) $\varepsilon \in [0, \frac{1}{\sqrt{MT}}]$, (ii) $\varepsilon \in (\frac{1}{\sqrt{MT}}, \frac{d}{\sqrt{T}})$, (iii) $\varepsilon \in [\frac{d}{\sqrt{T}}, +\infty]$?

We compare the performance of H-LINUCB with that of the following two algorithms: (a) Independent Learners (IND-OFUL), wherein each agent independently runs OFUL algorithm of Abbasi-Yadkori et al. [2011], and there is no communication between agents (zero collaboration), and (b) DISLINUCB, for which we use the implementation of Wang et al. [2020] without any modification.

**Simulation setup.**   We generate the $\varepsilon$-MALCB problem for $M = 60, d = 30, T = 10000$ via the following procedure. We first choose a value of $\varepsilon$ in each of the three dissimilarity regimes. Then we create the linear parameters $\{\theta_m\}_{m=1}^M$ as follows. Let $u, \{v_m\}_{m=1}^M$ be random vectors with unit norm. We set $\theta_m = c \cdot u + \frac{\varepsilon}{2} v_m$, where $c$ is a constant in the range $[0, 1 - \varepsilon]$. This guarantees $\|\theta_m\| \leq 1$ and $\|\theta_i - \theta_j\| \leq \varepsilon$ for any two agents $i, j$. At each round, for each agent, we create a new decision set with a size of $50$, each action is random and normalized to $1$. The random noise is sampled from the standard normal distribution, $\eta \sim \mathcal{N}(0, 1)$. We run each experiment 10 times, then report the

group regret averaged over the runs and the confidence intervals in Figure 1. Our code is available here: `https://github.com/anhddo/hlinUCB`.

**Results and discussions.** In regime (i), where the level of dissimilarity is small, plots (a) and (b) show that H-LINUCB retains a regret comparable with DISLINUCB.

In regime (ii), plots (c) and (d) illustrate the interpolation between the two extreme regimes.

In regime (iii), plots (e) and (f), DISLINUCB incurs linear regret, H-LINUCB has the same rate with IND-OFUL. This illustrates that collaboration brings no benefit when the dissimilarity is high.

## 6 Conclusions

In this paper, we studied the *heterogeneous* multi-agent linear contextual bandit problem. We formulated the problem under the notion of $\varepsilon$-MALCB, and provided the upper and lower bounds when $\varepsilon$ is known. We showed that our results are provably optimal in the regime where tasks are highly similar or highly dissimilar. Finally, we validated our theoretical results with numerical simulations on synthetic data.

A natural avenue for future work would be to close the gap in the regime $\varepsilon \in (\frac{1}{\sqrt{MT}}, \frac{d}{\sqrt{T}})$. Another research direction pertains to designing an adaptive algorithm when $\varepsilon$ is unknown. Such an algorithm would be practical and flexible enough to apply to a wide range of heterogeneous multi-agent bandit problems. We are also interested in extending this work to a more challenging setting such as Reinforcement Learning.

## Acknowledgements

This research was supported, in part, by DARPA GARD award HR00112020004, NSF CAREER award IIS-1943251, funding from the Institute for Assured Autonomy (IAA) at JHU, and the Spring'22 workshop on "Learning and Games" at the Simons Institute for the Theory of Computing.

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

# A Proof of Upper Bound

## A.1 Proof of Lemma 4.1

*Proof.* We prove that the true parameter of each agent lies in the *Confidence Ellipsoid* with high probability. In particular, we prove such results for two periods, for round $t < \tau$ and $t \geq \tau$. We denote $\delta_1, \delta_2$ as the probabilities such that true linear parameters lie outside the confidence ellipsoid for the period of $t < \tau$, and $t \geq \tau$, respectively.

For the round $t < \tau$, let $t_s$ be the last round that synchronization occurs, where $t_s < t$. Then each agent's statistics can be divided into two parts, its own data and the pooled data from other agents since round $t_s$. Specifically, for an agent $m$, at round $t$, we have the following for $V_{m,t}, b_{m,t}$,

$$V_{m,t} = \lambda I + \sum_{k=1}^{t-1} w_{m,k} x_{m,k} x_{m,k}^\top + \sum_{i \neq m} \sum_{k=1}^{t_s} w_{i,k} x_{i,k} x_{i,k}^\top,$$

$$b_{m,t} = \sum_{k=1}^{t-1} w_{m,k} x_{m,k} y_{m,k} + \sum_{i \neq m} \sum_{k=1}^{t_s} w_{i,k} x_{i,k} y_{i,k}.$$

We then have the following for $\hat{\theta}_{m,t}$,

$$
\begin{aligned}
\hat{\theta}_{m,t} &= V_{m,t}^{-1} b_{m,t} \\
&= V_{m,t}^{-1} \left[ \sum_{k=1}^{t-1} w_{m,k} x_{m,k} y_{m,k} + \sum_{i \neq m} \sum_{k=1}^{t_s} w_{i,k} x_{i,k} y_{i,k} \right] \\
&= V_{m,t}^{-1} \left[ \sum_{k=1}^{t-1} w_{m,k} x_{m,k} \left( x_{m,k}^\top \theta_m + \eta_{m,k} \right) + \sum_{i \neq m} \sum_{k=1}^{t_s} w_{i,k} x_{i,k} \left( x_{i,k}^\top \theta_i + \eta_{i,k} \right) \right] \\
&= V_{m,t}^{-1} \left[ \left( \sum_{k=1}^{t-1} w_{m,k} x_{m,k} x_{m,k}^\top + \sum_{i \neq m} \sum_{k=1}^{t_s} w_{i,k} x_{i,k} x_{i,k}^\top \right) \theta_m \right] \\
&\quad + V_{m,t}^{-1} \left( \sum_{i \neq m} \sum_{k=1}^{t_s} w_{i,k} x_{i,k} x_{i,k}^\top (\theta_i - \theta_m) \right) \\
&\quad + V_{m,t}^{-1} \left[ \sum_{k=1}^{t-1} w_{m,k} x_{m,k} \eta_{m,k} + \sum_{i \neq m} \sum_{k=1}^{t_s} w_{i,k} x_{i,k} \eta_{i,k} \right].
\end{aligned}
$$

Notice that the summation in the first term is the covariance matrix $V_{m,t}$ but missing the regularization term $\lambda I$; thus, the first term can be simplified as $\theta_m - \lambda V_{m,t}^{-1} \theta_m$. Applying triangle inequality, we have the following for $\left\| \hat{\theta}_{m,t} - \theta_m \right\|_{V_{m,t}}$,

$$
\left\| \hat{\theta}_{m,t} - \theta_m \right\|_{V_{m,t}} \leq \underbrace{\lambda \left\| \theta_m \right\|_{V_{m,t}^{-1}}}_{I_1 : \text{Regularization term}} + \underbrace{\left\| \sum_{i \neq m} \sum_{k=1}^{t_s} w_{i,k} x_{i,k} x_{i,k}^\top (\theta_i - \theta_m) \right\|_{V_{m,t}^{-1}}}_{I_2 : \text{Dissimilarity term}}
$$

$$
+ \underbrace{\left\| \sum_{k=1}^{t-1} w_{m,k} x_{m,k} \eta_{m,k} + \sum_{i \neq m} \sum_{k=1}^{t_s} w_{i,k} x_{i,k} \eta_{i,k} \right\|_{V_{m,t}^{-1}}}_{I_3 : \text{Noise term}}.
$$

For the regularization term $I_1$, it is upper bounded by,

$$
I_1 = \lambda \left\| \theta_m \right\|_{V_{m,t}^{-1}} \leq \frac{\lambda}{\lambda_{\min} \left( V_{m,t}^{1/2} \right)} \left\| \theta_m \right\|_2 \leq \sqrt{\lambda} \left\| \theta_m \right\|_2 \leq \sqrt{\lambda}.
$$

For the dissimilarity term $I_2$, we have,

$$
\begin{aligned}
I_2 &= \left\| \sum_{i \neq m} \sum_{k=1}^{t_s} w_{i,k} x_{i,k} x_{i,k}^\top (\theta_i - \theta_m) \right\|_{V_{m,t}^{-1}} \\
&\leq \sum_{i=1}^{M} \sum_{k=1}^{t_s} \left\| w_{i,k} x_{i,k} x_{i,k}^\top (\theta_i - \theta_m) \right\|_{V_{m,t}^{-1}} \\
&= \sum_{i=1}^{M} \sum_{k=1}^{t_s} w_{i,k} \left| x_{i,k}^\top (\theta_i - \theta_m) \right| \|x_{i,k}\|_{V_{m,t}^{-1}} \\
&\leq \sum_{i=1}^{M} \sum_{k=1}^{t_s} \|x_{i,k}\| \|\theta_i - \theta_m\| \alpha \\
&\leq \alpha \varepsilon M t_s,
\end{aligned}
$$

where we use triangle inequality in the first inequality, the second inequality holds since $w_{m,t} \leq \alpha / \|x_{m,t}\|_{V_{m,t}^{-1}}$ by the definition of $w_{m,t}$, the last inequality holds due to the bounded parameters assumption and the $\varepsilon$-MALCB definition (see Assumption 3.1 and Definition 3.1).

To bound term $I_3$, we denote vector $\bar{x}_{m,t} = \sqrt{w_{m,t}} x_{m,t}$, and the random noise $\bar{\eta}_{m,t} = \sqrt{w_{m,t}} \eta_{m,t}$. We have $w_{m,t} \leq 1$, therefore, $\bar{\eta}_{m,t}$ is 1-subGaussian. Furthermore, we rewrite the covariance matrix as $V_{m,t} = \lambda I + \sum_{s=1}^{t} \bar{x}_{m,s} \bar{x}_{m,s}^\top + \sum_{i \neq m} \sum_{k=1}^{t_s} \bar{x}_{i,k} \bar{x}_{i,k}^\top$. We then have the following for the noise term $I_3$,

$$
\begin{aligned}
I_3 &= \left\| \sum_{k=1}^{t-1} \bar{x}_{m,k} \bar{\eta}_{m,k} + \sum_{i \neq m} \sum_{k=1}^{t_s} \bar{x}_{i,k} \bar{\eta}_{i,k} \right\|_{V_{m,t}^{-1}} \\
&\leq \sqrt{2 \log \left( \frac{\det(V_{m,t})^{1/2} \det(V_0)^{-1/2}}{\delta_1} \right)} \\
&\leq \sqrt{d \log \left( \frac{1 + Mt/(\lambda d)}{\delta_1} \right)},
\end{aligned}
$$

where we apply Lemma C.4 in the first inequality. Now, putting these three terms together, we have the following *Confidence Ellipsoid* bound for round $t < \tau$,

$$
\left\| \hat{\theta}_{m,t} - \theta_m \right\|_{V_{m,t}} \leq \left( \sqrt{\lambda} + \alpha \varepsilon M t + \sqrt{d \log \left( \frac{1 + Mt/(\lambda d)}{\delta_1} \right)} \right).
$$

We now turn our attention to the round $t \geq \tau$. Since agents switch to *independent learning*, we follow the same argument for confidence ellipsoid as in Theorem 2 of Abbasi-Yadkori et al. [2011], and get

$$
\beta_t = \sqrt{\lambda} + \sqrt{d \log \left( \frac{1 + t/(\lambda d)}{\delta_2} \right)}.
$$

Finally, we complete the proof by taking a union bound over all agents. $\qquad\square$

## A.2 Proof of Theorem 4.1

Before proving Theorem 4.1, we give an upper bound on the pseudo-regret, $r_{m,t}$, in the next proposition. The proof follows standard arguments for linear bandits; most of the arguments can be extracted from Wang et al. [2020], Dubey and Pentland [2020].

**Proposition A.1.** *The pseudo-regret $r_t$ obtained by any agent $m$ at round $t$, is upper bounded as*

$$r_{m,t} \le 2\beta_t \min\left(1, \|x_{m,t}\|_{V_{m,t}^{-1}}\right).$$

*Proof.* Recall that an agent makes decisions optimistically: $(x_{m,t}, \tilde{\theta}_{m,t}) = \underset{(x,\theta) \in \mathcal{D}_{m,t} \times \mathcal{C}_{m,t}}{\arg\max} x^\top \theta$.

Let $x_{m,t}^\star$ denote the optimal action at round $t$ of agent $m$, i.e., $x_{m,t}^\star = \underset{x \in \mathcal{D}_{m,t}}{\arg\max} x^\top \theta_m$.

We then have the following for the pseudo-regret

$$
\begin{aligned}
r_{m,t} &= x_{m,t}^{\star}{}^\top \theta_m - x_{m,t}{}^\top \theta_m \\
&\le x_{m,t}{}^\top \tilde{\theta}_{m,t} - x_{m,t}{}^\top \theta_m && \text{(Since } (x_{m,t}, \tilde{\theta}_{m,t}) \text{ is optimistic)} \\
&= x_{m,t}{}^\top (\tilde{\theta}_{m,t} - \theta_m) \\
&= \langle V_{m,t}^{-1/2} x_{m,t}, V_{m,t}^{1/2}(\tilde{\theta}_{m,t} - \theta_m) \rangle && (V_{m,t} \succcurlyeq 0) \\
&\le \|x_{m,t}\|_{V_{m,t}^{-1}} \left\| \tilde{\theta}_{m,t} - \theta_m \right\|_{V_{m,t}} && \text{(Cauchy-Schwarz's inequality)} \\
&\le \|x_{m,t}\|_{V_{m,t}^{-1}} \left( \|\hat{\theta}_{m,t} - \tilde{\theta}_{m,t}\|_{V_{m,t}} + \|\hat{\theta}_{m,t} - \theta_m\|_{V_{m,t}} \right) && \text{(Triangle inequality)} \\
&\le 2\beta_t \|x_{m,t}\|_{V_{m,t}^{-1}} && \text{(Since } \theta_m, \tilde{\theta}_{m,t} \in \mathcal{C}_t) \\
&\le 2\beta_t \cdot \min\left(1, \|x_{m,t}\|_{V_{m,t}^{-1}}\right),
\end{aligned}
$$

where the last inequality is due to the fact that $\beta_t \ge 1$ and that suboptimality is no more than 2. $\square$

We now proceed to prove Theorem 4.1. The proof technique follows the analysis of group regret in Wang et al. [2020] and applies the arguments from Theorem 4.2 of He et al. [2022] to handle the dissimilarity.

**Theorem A.1** (Theorem 4.1 restated). *For a given $T$, and $\tau \le T$, if we set $D = \tau \log(M\tau)/(dM)$, and $\beta_t, \forall t \in [\tau]$ according to Lemma 4.1, then, given Assumption 3.1, with probability at least $1 - M\delta_1$, the group regret of Algorithm 1 up to round $\tau$, is upper bounded as*

$$
\begin{aligned}
\mathcal{R}(M, \tau) \le 4\sqrt{2}\Big( &\sqrt{\lambda d M \tau}(\xi_\tau)^{1.5} + \alpha\varepsilon\sqrt{d}(M\tau)^{1.5}\sqrt{\xi_\tau} + d\sqrt{M\tau}\xi_\tau + \frac{\sqrt{\lambda}d\xi_\tau}{\alpha} + \varepsilon dM\tau\xi_\tau \\
&+ \frac{(d\xi_\tau)^{1.5}}{\alpha} + \alpha\varepsilon\sqrt{d}(M\tau\xi_\tau)^{1.5} + d\sqrt{M\tau}\xi_\tau^2 \Big),
\end{aligned}
$$

*where $\xi_t = \log\left(\frac{1+Mt/(\lambda d)}{\delta_1}\right)$. Furthermore, if we choose $\lambda = 1, \alpha = \frac{\sqrt{d}}{\varepsilon M\tau}$, then the group regret is upper bounded as*

$$\mathcal{R}(M, \tau) \le 20\sqrt{2}\left(d\sqrt{M\tau}\xi_\tau^2 + \varepsilon dM\tau\xi_\tau^{1.5}\right).$$

*We refer to this group regret as the "collaboration" regret, and denote it as $\mathcal{R}_{\text{collab}}(M, T)$.*

*Proof.* Let $P$ be the total number of synchronization rounds, then, we can index the synchronization matrix $V_{syn}$ for $P$ epoch as $\{V_{syn,p}\}_{p=1}^P$, and define $V_p = \lambda I + V_{syn,p}$. Observe that $\det(V_0) = \det(\lambda I) = \lambda^d$ and $\det(V_P) \le (\text{trace}(V_P)/d)^d \le (\lambda + M\tau/d)^d$. Therefore,

$$\log \frac{\det(V_P)}{\det(V_0)} \le d\log\left(1 + \frac{M\tau}{\lambda d}\right). \tag{1}$$

By telescoping, we have that $\log \frac{\det(V_P)}{\det(V_0)} = \sum_{p=1}^P \log \frac{\det(V_p)}{\det(V_{p-1})}$. Therefore, we have at most $R = \lceil d\log(1 + \frac{M\tau}{\lambda d})\rceil$ epochs such that $\log \frac{\det(V_p)}{\det(V_{p-1})} \ge 1$; otherwise, it violates the condition

in Equation (1). WLOG, we use logarithm base 2 for the determinant ratio. This implies that for *all* but $R$ epochs,

$$1 \le \frac{\det(V_p)}{\det(V_{p-1})} \le 2, \tag{2}$$

We call the epochs satisfying Equation (2) as *good* epochs. We imagine a single agent playing $M(\tau - 1)$ actions $x_{1,1}, x_{2,1}, \ldots, x_{M-1,\tau-1}, x_{M,\tau-1}$ sequentially. Let $W_{m,t} = \lambda I + \sum_{i=1}^{M} \sum_{s=1}^{t-1} w_{i,s} x_{i,s} x_{i,s}^\top + \sum_{j=1}^{m-1} w_{j,t} x_{j,t} x_{j,t}^\top$ be the covariance matrix of this imaginary agent when it gets to $x_{m,t}$. If $x_{m,t}$ belongs to a good epoch, say the $j$-th epoch, we have the following:

$$1 \le \frac{\|x_{m,t}\|_{V_{m,t}^{-1}}}{\|x_{m,t}\|_{W_{m,t}^{-1}}} \le \sqrt{\frac{\det(W_{m,t})}{\det(V_{m,t})}} \le \sqrt{\frac{\det(V_j)}{\det(V_{j-1})}} \le \sqrt{2}, \tag{3}$$

where the first inequality is due to the fact that $W_{m,t} \succcurlyeq V_{m,t}$, and the second inequality follows from Lemma C.2.

Now, applying the Proposition A.1, we bound the pseudo-regret $r_{m,t}$ of these good epochs as follows:

$$
\begin{aligned}
r_{m,t} &\le 2\beta_t \min\left(1, \|x_{m,t}\|_{V_{m,t}^{-1}}\right) \\
&\le 2\beta_t \min\left(1, \|x_{m,t}\|_{W_{m,t}^{-1}} \sqrt{\frac{\det(W_{m,t})}{\det(V_{m,t})}}\right) \\
&\le 2\sqrt{2}\beta_t \min\left(1, \|x_{m,t}\|_{W_{m,t}^{-1}}\right),
\end{aligned}
$$

where we use Lemma C.2 in the second inequality, and Equation (3) in the third inequality.

Let $\mathcal{R}_{\text{good}}(M, \tau)$ be the group regret of these good epochs up to round $\tau$. Suppose $P_{\text{good}}$ contains all the good epochs, and $\mathcal{B}_p$ contains all the pairs $(m, t)$ belonging to epoch $p$. We have

$$
\begin{aligned}
\mathcal{R}_{\text{good}}(M, \tau) &= \sum_{p \in P_{\text{good}}} \sum_{(m,t) \in \mathcal{B}_p} r_{m,t} \\
&\le \sum_{p \in P_{\text{good}}} \sum_{(m,t) \in \mathcal{B}_p} 2\sqrt{2}\beta_{\tau-1} \min\left(1, \|x_{m,t}\|_{W_{m,t}^{-1}}\right) \\
&= \underbrace{\sum_{p \in P_{\text{good}}} \sum_{(m,t) \in \mathcal{B}_p \wedge w_{m,t}=1} 2\sqrt{2}\beta_{\tau-1} \min\left(1, \|x_{m,t}\|_{W_{m,t}^{-1}}\right)}_{I_1:\text{Rounds of good epoch with } w_{m,t}=1} \\
&\quad + \underbrace{\sum_{p \in P_{\text{good}}} \sum_{(m,t) \in \mathcal{B}_p \wedge w_{m,t}<1} 2\sqrt{2}\beta_{\tau-1} \min\left(1, \|x_{m,t}\|_{W_{m,t}^{-1}}\right)}_{I_2:\text{Rounds of good epoch with } w_{m,t}<1},
\end{aligned}
$$

where in the last equality we split the summation to two cases: $w_{m,t} = 1$, and $w_{m,t} < 1$.

For term $I_1$, we consider all the pair $(m, t)$ in good epochs up to round $\tau$ such that $w_{m,t} = 1$; we assume that we have a total of such $K$ pairs, and these pairs can be *sequentially* listed as $\{(\bar{m}_1, \bar{t}_1), (\bar{m}_2, \bar{t}_2), \cdots, (\bar{m}_K, \bar{t}_K)\}$. With this notation, for $i \le K$, we construct the auxiliary covariance matrix $A_{\bar{m}_i, \bar{t}_i} = \lambda I + \sum_{k=1}^{i-1} x_{\bar{m}_k, \bar{t}_k} x_{\bar{m}_k, \bar{t}_k}^\top$. Thus, we have $W_{\bar{m}_i, \bar{t}_i} \succcurlyeq A_{\bar{m}_i, \bar{t}_i}$, which implies $\|x_{\bar{m}_i, \bar{t}_i}\|_{W_{\bar{m}_i, \bar{t}_i}^{-1}} \le \|x_{\bar{m}_i, \bar{t}_i}\|_{A_{\bar{m}_i, \bar{t}_i}^{-1}}$.

We now have the following for term $I_1$:

$$
\begin{aligned}
I_1 &= \sum_{p \in P_{\text{good}}} \sum_{(m,t) \in \mathcal{B}_p \wedge w_{m,t}=1} 2\sqrt{2}\beta_{\tau-1} \min\left(1, \|x_{m,t}\|_{W_{m,t}^{-1}}\right) \\
&= \sum_{k=1}^{K} 2\sqrt{2}\beta_{\tau-1} \min\left(1, \|x_{\bar{m}_k,\bar{t}_k}\|_{W_{\bar{m}_k,\bar{t}_k}^{-1}}\right) \\
&\leq \sum_{k=1}^{K} 2\sqrt{2}\beta_{\tau-1} \min\left(1, \|x_{\bar{m}_k,\bar{t}_k}\|_{A_{\bar{m}_k,\bar{t}_k}^{-1}}\right) \\
&\leq 2\sqrt{2}\beta_{\tau-1}\sqrt{M\tau}\sqrt{\sum_{i=1}^{K} \min\left(1, \|x_{\bar{m}_k,\bar{t}_k}\|_{A_{\bar{m}_k,\bar{t}_k}^{-1}}^2\right)} \\
&\leq 4\beta_{\tau-1}\sqrt{2M\tau \log\left(\frac{\det(V_P)}{\det(V_0)}\right)} \\
&\leq 4\sqrt{2}\beta_{\tau-1}\sqrt{dM\tau \log\left(1 + \frac{M\tau}{\lambda d}\right)} \\
&\leq 4\sqrt{2}\beta_{\tau-1}\sqrt{dM\tau\xi_\tau},
\end{aligned}
$$

where in the second inequality we use Cauchy Schwarz's inequality and the fact that $K \leq M\tau$, and use Lemma C.3 in the second inequality.

For term $I_2$, since we consider the case such that $w_{m,t} < 1$, thus, by the definition of $w_{m,t}$ we have $\frac{1}{\alpha}w_{m,t} \cdot \|x_{m,t}\|_{V_{m,t}^{-1}} = 1$. Therefore, we have,

$$
\begin{aligned}
I_2 &= 2\sqrt{2} \sum_{p \in P_{\text{good}}} \sum_{(m,t) \in \mathcal{B}_p \wedge w_{m,t}<1} \min\left(1, \beta_\tau \|x_{m,t}\|_{W_{m,t}^{-1}}\right) \\
&= 2\sqrt{2} \sum_{p \in P_{\text{good}}} \sum_{(m,t) \in \mathcal{B}_p \wedge w_{m,t}<1} \min\left(1, \beta_\tau \frac{w_{m,t}\|x_{m,t}\|_{V_{m,t}^{-1}}}{\alpha} \|x_{m,t}\|_{W_{m,t}^{-1}}\right) \\
&\leq 4 \sum_{p \in P_{\text{good}}} \sum_{(m,t) \in \mathcal{B}_p \wedge w_{m,t}<1} \min\left(1, \beta_\tau \frac{w_{m,t}\|x_{m,t}\|_{W_{m,t}^{-1}}^2}{\alpha}\right),
\end{aligned}
$$

where the last inequality follows from Equation (3).

Similarly to what we have done with $I_1$, we consider all the pair $(m,t)$ in good epochs up to round $\tau$ such that $w_{m,t} < 1$; we assume that we have a total of such $K$ pairs, and these pairs can be *sequentially* listed as $\{(\bar{m}_1, \bar{t}_1), (\bar{m}_2, \bar{t}_2), \cdots, (\bar{m}_K, \bar{t}_K)\}$. Furthermore, we define $\bar{x}_{m,t} = \sqrt{w_{m,t}}x_{m,t}$, and construct the auxiliary covariance matrix,

$$
\bar{W}_{\bar{m}_i,\bar{t}_i} = \lambda I + \sum_{k=1}^{i-1} w_{\bar{m}_k,\bar{t}_k} x_{\bar{m}_k,\bar{t}_k} x_{\bar{m}_k,\bar{t}_k}^\top = \lambda I + \bar{x}_{\bar{m}_k,\bar{t}_k} \bar{x}_{\bar{m}_k,\bar{t}_k}^\top.
$$

Thus, we have $W_{\bar{m}_i,\bar{t}_i} \succcurlyeq \bar{W}_{\bar{m}_i,\bar{t}_i}$, which implies $\|x_{\bar{m}_i,\bar{t}_i}\|_{W_{\bar{m}_i,\bar{t}_i}^{-1}} \leq \|x_{\bar{m}_i,\bar{t}_i}\|_{\bar{W}_{\bar{m}_i,\bar{t}_i}^{-1}}$.

Therefore,

$$
\begin{aligned}
I_2 &\le 4 \sum_{k=1}^{K} \min\left(1, \beta_{\tau-1} w_{\bar{m}_k, \bar{t}_k}/\alpha \left\| x_{\bar{m}_k, \bar{t}_k} \right\|^2_{W^{-1}_{\bar{m}_k, \bar{t}_k}}\right) \\
&\le 4 \sum_{k=1}^{K} \min\left(1 + \beta_{\tau-1}/\alpha, (1 + \beta_{\tau-1}/\alpha)\| \sqrt{w_{\bar{m}_k, \bar{t}_k}} x_{\bar{m}_k, \bar{t}_k} \|^2_{W^{-1}_{\bar{m}_k, \bar{t}_k}}\right) \\
&\le 4(1 + \beta_{\tau-1}/\alpha) \sum_{k=1}^{K} \min\left(1, \|\bar{x}_{\bar{m}_k, \bar{t}_k}\|^2_{\bar{W}^{-1}_{\bar{m}_k, \bar{t}_k}}\right) \\
&\le 4(1 + \beta_{\tau-1}/\alpha) d \log\left(1 + \frac{M\tau}{\lambda d}\right) \\
&\le 4(1 + \beta_{\tau-1}/\alpha) d\xi_\tau,
\end{aligned}
$$

where we use Lemma C.3 in the fourth inequality. Putting the bounds for $I_1$ and $I_2$ together, we have

$$
\begin{aligned}
\mathcal{R}_{\mathrm{good}}(M, \tau) = I_1 + I_2 &\le 4\sqrt{2}\beta_{\tau-1}\sqrt{dM\tau\xi_\tau} + 4\left(1 + \frac{\beta_{\tau-1}}{\alpha}\right) d\xi_\tau \\
&= 4\sqrt{2}\left(\beta_{\tau-1}\sqrt{dM\tau\xi_\tau} + d\xi_\tau + \frac{\beta_{\tau-1}}{\alpha} d\xi_\tau\right).
\end{aligned}
$$

Plugging $\beta_{\tau-1} = \sqrt{\lambda} + \alpha\varepsilon M\tau + \sqrt{d\log(\frac{1+M\tau/\lambda}{\delta_1})}$, the regret of the good epochs is bounded as

$$
\mathcal{R}_{\mathrm{good}}(M, \tau) \le 4\sqrt{2}\left(\sqrt{\lambda dM\tau\xi_\tau} + \alpha\varepsilon\sqrt{d}(M\tau)^{1.5}\sqrt{\xi_\tau} + d\sqrt{M\tau}\xi_\tau + \frac{\sqrt{\lambda}d\xi_\tau}{\alpha} + \varepsilon dM\tau\xi_\tau + \frac{(d\xi_\tau)^{1.5}}{\alpha}\right).
$$

Now, we focus on the epochs that are *not good*, which do not satisfy Equation (2). Let $p$ be one such epoch, and let $t_0$ be the first round and $n$ be the length, respectively, of the epoch $p$. Recall that at the beginning of each epoch, agents' covariance matrices are synchronized, i.e., $V_{m, t_0} = \lambda I + V_{syn}, \forall m \in [M]$. We can then bound the regret of the (bad) epoch $p$ as follows.

$$
\begin{aligned}
\mathcal{R}_{\mathrm{bad}}(M, p) &\le 2\beta_{\tau-1} \sum_{m=1}^{M} \sum_{t=t_0}^{t_0+n} \min\left(1, \|x_{m,t}\|_{V^{-1}_{m,t}}\right) \\
&\le 2\beta_{\tau-1} \sum_{m=1}^{M} \sqrt{n} \sqrt{\sum_{t=t_0}^{t_0+n} \min\left(1, \|x_{m,t}\|^2_{V^{-1}_{m,t}}\right)} \\
&\le 2\beta_{\tau-1} \sum_{m=1}^{M} \sqrt{n \log \frac{\det(V_{m, t_0+n})}{\det(V_{m, t_0})}},
\end{aligned}
$$

where we use Cauchy-Schwarz's inequality and Lemma C.3 in the second inequality.

Lets say the synchronization is triggered on round $t_0 + n + 1$, i.e., for one of the agents we have that $(n+1)\log\frac{\det(V_{m, t_0+n+1})}{\det(V_{m, t_0})} > D$. Since $n\log\frac{\det(V_{m, t_0+n})}{\det(V_{m, t_0})} < D$ for all $m \in [M]$, for this bad epoch (i.e., from round $t_0$ to round $t_0 + n$), we can bound the group regret as

$$
\mathcal{R}_{\mathrm{bad}}(M, p) \le 2\beta_{\tau-1} M\sqrt{D}.
$$

As we argued earlier, since $\det V_p \le (\lambda + \frac{MT}{d})^d$, the bad epochs are rare, i.e., there are at most $R = \lceil d \log(1 + \frac{M\tau}{\lambda d}) \rceil$ (otherwise, if $\det(V_p)/\det(V_{p-1}) > 2$ for more than $R$ rounds, then the condition of Equation (1) is violated). Setting $D = \tau \log(M\tau)/(dM)$ and plugging $\beta_{\tau-1}$ into the bound above,

$$
\begin{aligned}
\mathcal{R}_{\mathrm{bad}}(M, \tau) &\le 2R\beta_{\tau-1} M\sqrt{D} \\
&\le 2\left(d\xi_\tau\left(\sqrt{\lambda} + \alpha\varepsilon M\tau + \sqrt{d\xi_\tau}\right) M\sqrt{D}\right) \\
&\le 2\left(\sqrt{\lambda dM\tau}(\xi_\tau)^{1.5} + \alpha\varepsilon\sqrt{d}(M\tau\xi_\tau)^{1.5} + d\sqrt{M\tau}\xi_\tau^2\right).
\end{aligned}
$$

We finish the proof by putting the regret of the good and the bad epochs together,

$$\mathcal{R}(M, \tau) = \mathcal{R}_{\text{good}}(M, \tau) + \mathcal{R}_{\text{bad}}(M, \tau)$$

$$\leq 4\sqrt{2}\Big(\sqrt{\lambda d M \tau}(\xi_\tau)^{1.5} + \alpha\varepsilon\sqrt{d}(M\tau)^{1.5}\sqrt{\xi_\tau} + d\sqrt{M\tau}\xi_\tau + \frac{\sqrt{\lambda}d\xi_\tau}{\alpha} + \varepsilon d M \tau \xi_\tau$$

$$+ \frac{(d\xi_\tau)^{1.5}}{\alpha} + \alpha\varepsilon\sqrt{d}(M\tau\xi_\tau)^{1.5} + d\sqrt{M\tau}\xi_\tau^2\Big).$$

□

## A.3 Proof of Theorem 4.2

*Proof.* Setting $\delta_1 = \delta_2 = 1/(M^2 T)$, then the group regret caused by failure event, which does not satisfy Lemma 4.1, is at most $MT \cdot (M\delta_1 + M\delta_2) = \mathcal{O}(1)$. Thus, we mainly consider the group regret when Lemma 4.1 holds.

From round $\tau$ onward, agents switch to *independent learning* mode, this is the group regret from round $\tau$ to the end, we denote it as $\mathcal{R}_{\text{ind}}(M, T)$. Furthermore, for round $t \geq \tau$, we have $V_{m,t} = \lambda I + \sum_{s=\tau}^{t-1} x_{m,s} x_{m,s}^\top$ as the gram matrix which only contains agent's data.

Applying Proposition A.1, we have the following

$$\mathcal{R}_{\text{ind}}(M, T) \leq \sum_{m=1}^{M} \sum_{t=\tau}^{T} 2\beta_t \min\left(1, \|x_{m,t}\|_{V_{m,t}^{-1}}\right)$$

$$\leq 2\sum_{m=1}^{M} \beta_T \sqrt{T - \tau} \sqrt{\sum_{t=\tau}^{T} \min\left(1, \|x_{m,t}\|_{V_{m,t}^{-1}}^2\right)}$$

$$\leq 2\sum_{m=1}^{M} \left(\sqrt{\lambda} + \sqrt{d\log\left(\frac{1 + T/(\lambda d)}{\delta_2}\right)}\right)\sqrt{T - \tau}\sqrt{2d\log\left(1 + \frac{T}{\lambda d}\right)}$$

$$\leq 4\sqrt{2}Md\sqrt{T - \tau}\xi_T,$$

where we use Cauchy-Schwarz's inequality in the first inequality, and Lemma 4.1 in the second inequality. Combining with the regret from the beginning up to round $\tau$ from Theorem 4.1, we have

$$\mathcal{R}(M, T) = \mathcal{R}_{\text{collab}}(M, T) + \mathcal{R}_{\text{ind}}(M, T)$$
$$\leq 20\sqrt{2}\left(d\sqrt{M\tau} + \varepsilon d M \tau + dM\sqrt{T - \tau}\right)\xi_T^2 \tag{4}$$

We notice that Equation (4) has the second term that is linear in term of $M\tau$. To avoid linear regret, the choice of $\tau$ needs to adapt to the dissimilarity level $\varepsilon$.

For the last two terms, by Cauchy-Schwarz's inequality, we have $\varepsilon d M \tau + dM\sqrt{T - \tau} \leq dM\sqrt{2(\varepsilon^2\tau^2 + T - \tau)}$, this term can be optimized by setting $\tau = \frac{1}{2\varepsilon^2}$. We need to restrict $\tau = \min(\lfloor\frac{1}{2}\varepsilon^{-2}\rfloor, T)$ since $\tau$ can not exceed $T$. In other words, all agents can collaborate up to round $\min\left(\lfloor\frac{1}{2}\varepsilon^{-2}\rfloor, T\right)$.

Next, we consider the following two cases:

Case 1: For $\frac{1}{2}\varepsilon^{-2} \leq T$, we have that $dM\sqrt{T - \tau} \leq dM\sqrt{T}$, and $\varepsilon d M \tau = \frac{dM}{2\varepsilon} \leq dM\sqrt{T}$. Thus, $\varepsilon d M \tau + dM\sqrt{T - \tau} \leq 2dM\sqrt{T}$.

Case 2: For $\frac{1}{2}\varepsilon^{-2} > T$, we have $\tau = T$. This implies that $\varepsilon d M \tau + dM\sqrt{T - \tau} = \varepsilon d M T$. We also have $\varepsilon d M T < \frac{1}{\sqrt{2}}dM\sqrt{T}$ due to $\frac{1}{2}\varepsilon^2 > T$.

Notice that, in both cases, $\varepsilon d M \tau + dM\sqrt{T - \tau}$ is evaluated as small as $\varepsilon d M T$ and always be upper bounded by $2dM\sqrt{T}$. Therefore, in Equation (4), the summation of the last two terms of the group regret is upper bounded by $2\min\{\varepsilon d M T, dM\sqrt{T}\}$.

By the choice of $\delta_1, \delta_2, \lambda$, and assume that $d \geq 1$, we can upper bound $\xi_T \leq 4\log(MT)$. Therefore, we have the following for the expected group regret

$$\mathbb{E}\left[\mathcal{R}(M,T)\right] \leq 20\sqrt{2}\left(d\sqrt{MT} + 2\min\{\varepsilon dMT, dM\sqrt{T}\}\right)\xi_T^2$$

$$\leq 320\sqrt{2}\left(d\sqrt{MT} + 2\min\{\varepsilon dMT, dM\sqrt{T}\}\right)\log^2(MT)$$

$\square$

# B  Proof of Lower Bound

## B.1  Proof of Theorem 4.3

Before proving the theorem, we give a formal definition of $\varepsilon$-MALCB problem for $p$-norm,

**Definition B.1.** *Given an instance of multi-agent linear bandits, it belongs to the class of $p$-norm $\varepsilon$-MALCB, which we denote as $\mathcal{I}_p(\varepsilon)$, if for all $i, j \in [M]$, $\|\theta_i - \theta_j\|_p \leq \varepsilon$.*

To prove Theorem 4.3, we use the results of the two following lemmas on *max-norm $\varepsilon$-MALCB* instances.

**Lemma B.1.** *Let $\mathcal{I}_\infty(\varepsilon)$ be the class of max-norm $\varepsilon$-MALCB instances that satisfy the Assumption 3.1. For $\varepsilon \geq 0$, we have the following,*

$$\inf_{\mathcal{A}} \sup_{\mathcal{I} \in \mathcal{I}_\infty(\varepsilon)} \mathcal{R}_{\mathcal{A},\mathcal{I}}(M,T) = \Omega(d\sqrt{MT}).$$

**Lemma B.2.** *Let $\mathcal{I}_\infty(\varepsilon)$ be the class of max-norm $\varepsilon$-MALCB instances that satisfy the Assumption 3.1. Assume $d \leq 2T, \frac{d^2}{48} \leq T$. For $\varepsilon \geq 0$, we have the following,*

$$\inf_{\mathcal{A}} \sup_{\mathcal{I} \in \mathcal{I}_\infty(\varepsilon)} \mathcal{R}_{\mathcal{A},\mathcal{I}}(M,T) = \Omega\left(\min\left\{\varepsilon MT\sqrt{d}, dM\sqrt{T}\right\}\right).$$

*Proof of Theorem 4.3.* Combining the results of Lemma B.1 and Lemma B.2, we have the following results for the class of max-norm $\varepsilon$-MALCB, for $\varepsilon \geq 0$,

$$\inf_{\mathcal{A}} \sup_{\mathcal{I} \in \mathcal{I}_\infty(\varepsilon)} \mathcal{R}_{\mathcal{A},\mathcal{I}}(M,T) = \Omega\left(d\sqrt{MT} + \min\left\{\varepsilon MT\sqrt{d}, dM\sqrt{T}\right\}\right).$$

In addition, $\|x\|_\infty \leq \varepsilon$ implies $\|x\|_2 \leq \varepsilon\sqrt{d}$, therefore, $\mathcal{I}_\infty(\varepsilon) \subseteq \mathcal{I}_2(\varepsilon\sqrt{d})$. In other words, for $\varepsilon \geq 0$,

$$\inf_{\mathcal{A}} \sup_{\mathcal{I} \in \mathcal{I}_2(\varepsilon)} \mathcal{R}_{\mathcal{A},\mathcal{I}}(M,T) \geq \inf_{\mathcal{A}} \sup_{\mathcal{I} \in \mathcal{I}_\infty\left(\frac{\varepsilon}{\sqrt{d}}\right)} \mathcal{R}_{\mathcal{A},\mathcal{I}}(M,T) = \Omega\left(d\sqrt{MT} + \min\left\{\varepsilon MT, dM\sqrt{T}\right\}\right).$$

$\square$

*Proof of Lemma B.1.* In this lemma, we prove that solving any $\varepsilon$-MALCB instance for $T$ rounds is at least as hard as solving a (single-agent) linear bandits for $MT$ rounds. We prove the lemma by contradiction, which is based on linear bandits lower bound of Lemma C.1. We have that $\mathcal{I}_\infty(\varepsilon') \subseteq \mathcal{I}_\infty(\varepsilon)$, for $0 \leq \varepsilon' \leq \varepsilon$. This implies, for $\varepsilon \geq 0$,

$$\inf_{\mathcal{A}} \sup_{\mathcal{I} \in \mathcal{I}_\infty(\varepsilon)} \mathcal{R}_{\mathcal{A},\mathcal{I}}(M,T) \geq \inf_{\mathcal{A}} \sup_{\mathcal{I} \in \mathcal{I}_\infty(0)} \mathcal{R}_{\mathcal{A},\mathcal{I}}(M,T).$$

We completes the proof by proving $\inf_{\mathcal{A}} \sup_{\mathcal{I} \in \mathcal{I}_\infty(0)} \mathcal{R}_{\mathcal{A},\mathcal{I}}(M,T) = \Omega\left(d\sqrt{MT}\right)$.

Now, we assume there exists an algorithm $\mathcal{A}$ which achieves $\sup_{\mathcal{I} \in \mathcal{I}_\infty(0)} \mathcal{R}_{\mathcal{A},\mathcal{I}}(M,T) < \frac{d\sqrt{MT}}{16\sqrt{3}}$.

We observe that $\mathcal{I}_\infty(0)$ is the class of multi-agent solving exactly the same linear bandits problem. We simulate the algorithm $\mathcal{B}$ on a single agent linear bandits for $MT$ rounds by the protocol of $M$ agents solving an identical linear bandits for $T$ rounds. Therefore, if we have $\mathcal{R}_{\mathcal{A},\mathcal{I}}(M,T) < \frac{d\sqrt{MT}}{16\sqrt{3}}$ then we also achieve $\mathcal{R}_{\mathcal{B},\mathcal{I}}(MT) < \frac{d\sqrt{MT}}{16\sqrt{3}}$, which contradicts Lemma C.1. Thus, we have $\sup_{\mathcal{I} \in \mathcal{I}_\infty(0)} \mathcal{R}_{\mathcal{A},\mathcal{I}}(M,T) \geq \frac{d\sqrt{MT}}{16\sqrt{3}}$, which completes the proof. $\square$

*Proof of Lemma B.2.* We extend Lemma C.1 to multi-agent linear contextual bandit by constructing the following *max-norm $\varepsilon$-MALCB* instance.

**Max-norm $\varepsilon$-MALCB instance.** The set of linear parameter of $M$ agents $\{\theta_m\}_{m=1}^M$ belong to a $d$-dimensional hypercube $\theta_m \in \{\pm\varepsilon\}^d$, where $\varepsilon \in [0, \frac{1}{\sqrt{d}}]$. Let $\mathcal{D} = \{x \in \mathbb{R}^d : \|x\|_2 \leq 1\}$ be the action set given to agents at every round. The reward when agent $m$ picks action $x$ is defined as $r_{m,x} = \theta_m^\top x + \eta_{m,x}$, where the noise samples from a standard normal distribution, $\eta_{m,x} \sim \mathcal{N}(0, 1)$.

We first verify if the instance satisfies the Assumption 3.1. It satisfies the context assumption since we also restrict the context vector to lie in the unit ball. It also satisfies the $\|\theta_m\| \leq 1$ assumption because we restrict $\varepsilon \in [0, \frac{1}{\sqrt{d}}]$. Finally, $\eta_{m,x} \sim \mathcal{N}(0, 1)$ is 1-subGaussian distribution. This instance belongs to $\mathcal{I}_\infty(2\varepsilon)$ since $\|\theta_i - \theta_j\|_\infty \leq 2\varepsilon$ for all $i, j \in [M]$.

Now, we proceed to prove Lemma B.2. Let $\{\theta_m\}_{m=1}^M$ be a set of parameters of the max-norm $\varepsilon$-MALCB instance. For brevity, we omit the commas in the subscripts when it is clear from the context, e.g., $\theta_{mi} = \theta_{m,i}$ or $x_{tmi} = x_{t,m,i}, \forall t \in [T], m \in [M], i \in [d]$. Given $m \in [M], i \in [d]$, we define the stopping time $\tau_{mi} = T \wedge \min\{t : \sum_{s=1}^t x_{smi}^2 \geq t/d\}$, and the function $U_{mi}(z) = \sum_{t=1}^{\tau_{mi}}(\frac{1}{\sqrt{d}} - x_{tmi} \cdot z)^2$. We then have following result for $U_{mi}(1)$:

$$U_{mi}(1) = \sum_{t=1}^{\tau_{mi}} \left(\frac{1}{\sqrt{d}} - x_{tmi}\right)^2 \leq 2\sum_{t=1}^{\tau_{mi}} \frac{1}{d} + 2\sum_{t=1}^{\tau_{mi}} x_{tmi}^2 \leq \frac{4T}{d} + 2. \tag{5}$$

Let $x_m^\star$ be the optimal action of agent $m$. We then have the following for the group regret

$$\mathcal{R}_{\mathcal{A}}(M, T) = \mathbb{E}_{\{\theta_m\}_{m=1}^M}\left[\sum_{t=1}^T \sum_{m=1}^M \sum_{i=1}^d (x_{mi}^\star \theta_{mi} - x_{tmi}\theta_{mi})\right]$$

$$= \varepsilon \cdot \mathbb{E}_{\{\theta_m\}_{m=1}^M}\left[\sum_{t=1}^T \sum_{m=1}^M \sum_{i=1}^d \left(\frac{1}{\sqrt{d}} - x_{tmi}\,\mathrm{sign}(\theta_{mi})\right)\right]$$

$$\geq \frac{\varepsilon\sqrt{d}}{2} \cdot \mathbb{E}_{\{\theta_m\}_{m=1}^M}\left[\sum_{t=1}^T \sum_{m=1}^M \sum_{i=1}^d \left(\frac{1}{\sqrt{d}} - x_{tmi}\,\mathrm{sign}(\theta_{mi})\right)^2\right]$$

$$\geq \frac{\varepsilon\sqrt{d}}{2} \cdot \sum_{m=1}^M \sum_{i=1}^d \mathbb{E}_{\{\theta_m\}_{m=1}^M}\left[\sum_{t=1}^{\tau_{mi}} \left(\frac{1}{\sqrt{d}} - x_{tmi}\,\mathrm{sign}(\theta_{mi})\right)^2\right],$$

where we use the fact that optimal action $x_m^\star = [\frac{\theta_{m1}}{\|\theta_m\|}, \cdots, \frac{\theta_{md}}{\|\theta_m\|}]^\top$, which is a unit vector and has the same direction with $\theta_m$, the first inequality holds due to $\|x_{tm}\|^2 \leq 1$.

Now, let $\{\theta_m'\}_{m=1}^M$ be another set of linear parameters that are different at only *one coordinate* of the linear parameter of only *one agent* compared to $\{\theta_m\}_{m=1}^M$. Specifically, fix $g \in [M], i \in [d]$, we have $\theta_{g,i} = -\theta_{g,i}'$; otherwise, $\theta_{k,j} = \theta_{k,j}'$, for $(k, j) \neq (g, i), \forall k \in [M], j \in [d]$. To simplify the notion, we define $\Phi, \Phi'$ as $Md$-dimensional vectors, where $\Phi, \Phi' \in \{\pm\varepsilon\}^{Md}$. That is, $\Phi, \Phi'$ represent $\{\theta_m\}_{m=1}^M, \{\theta_m'\}_{m=1}^M$, respectively. We let $\mathbb{P}$ and $\mathbb{P}'$ be the law of $U_{gi}(z)$ w.r.t. the interaction of the multi-agent linear bandits induced by $\Phi, \Phi'$, respectively. We then get

$$\mathbb{E}_\Phi[U_{gi}(1)] \geq \mathbb{E}_{\Phi'}[U_{gi}(1)] - \left(\frac{4T}{d} + 2\right)\sqrt{\frac{1}{2}D_{KL}(\mathbb{P}\|\mathbb{P}')}$$

$$\geq \mathbb{E}_{\Phi'}[U_{gi}(1)] - \frac{\varepsilon}{2}\left(\frac{4T}{d} + 2\right)\sqrt{\mathbb{E}\left[\sum_{t=1}^{\tau_{gi}} x_{tgi}^2\right]}$$

$$\geq \mathbb{E}_{\Phi'}[U_{gi}(1)] - \frac{\varepsilon}{2}\left(\frac{4T}{d} + 2\right)\sqrt{\frac{T}{d} + 1}$$

$$\geq \mathbb{E}_{\Phi'}[U_{gi}(1)] - 4\sqrt{3}\varepsilon\left(\frac{T}{d}\right)^{3/2},$$

where the first inequality holds due to Lemma C.5 and Equation (5), the second inequality follows the stopping time arguments from Lattimore and Csaba [2020], the last inequality holds due to the assumption that $d \leq 2T$. Then,

$$\mathbb{E}_\Phi \left[ U_{gi}(1) \right] + \mathbb{E}_{\Phi'} \left[ U_{gi}(-1) \right] \geq \mathbb{E}_{\Phi'} \left[ U_{gi}(1) + U_{gi}(-1) \right] - 4\sqrt{3}\varepsilon \left( \frac{T}{d} \right)^{3/2}$$

$$= 2\mathbb{E}_{\Phi'} \left[ \frac{\tau_{gi}}{d} + \sum_{t=1}^{\tau_{gi}} x_{tgi}^2 \right] - 4\sqrt{3}\varepsilon \left( \frac{T}{d} \right)^{3/2}$$

$$\geq \frac{2T}{d} - 4\sqrt{3}\varepsilon \left( \frac{T}{d} \right)^{3/2} \geq \frac{T}{d},$$

where the last inequality holds for $\varepsilon \in [0, \frac{1}{4\sqrt{3}}\sqrt{\frac{d}{T}}]$, this satisfies the requirement $\varepsilon \in [0, \frac{1}{\sqrt{d}}]$ of the instance construction due to the assumption that $\frac{d^2}{48} \leq T$. Furthermore, we denote $\Phi_{-mi}$ as a $(Md-1)$-dimensional vector, which is obtained by excluding $\Phi_{mi}$ from vector $\Phi$. Let $\mathcal{R}_\Phi(M, T)$ be the regret w.r.t. $\Phi$, and applying *randomization hammer*, we have the following,

$$\sum_{\Phi \in \{\pm\varepsilon\}^{Md}} \mathcal{R}_\Phi(M, T) \geq \frac{\varepsilon\sqrt{d}}{2} \sum_{m=1}^{M} \sum_{i=1}^{d} \sum_{\Phi \in \{\pm\varepsilon\}^{Md}} \mathbb{E}_\Phi \left[ U_{mi} \left( \text{sign} \left( \varepsilon_{mi} \right) \right) \right]$$

$$= \frac{\varepsilon\sqrt{d}}{2} \sum_{m=1}^{M} \sum_{i=1}^{d} \sum_{\Phi_{-mi} \in \{\pm\varepsilon\}^{Md-1}} \sum_{\Phi_{mi} \in \{\pm\varepsilon\}} \mathbb{E}_\Phi \left[ U_{mi} \left( \text{sign} \left( \varepsilon_{mi} \right) \right) \right]$$

$$\geq \frac{\varepsilon\sqrt{d}}{2} \sum_{m=1}^{M} \sum_{i=1}^{d} \sum_{\Phi_{-mi} \in \{\pm\varepsilon\}^{Md-1}} \frac{T}{d} = 2^{Md-2}\varepsilon MT\sqrt{d}.$$

Therefore, we conclude that there exists an instance with the parameter set of $\{\theta_m\}_{m=1}^{M}$ such that $\mathcal{R}(M, T) \geq \frac{\varepsilon MT\sqrt{d}}{4}$, for $\varepsilon \in [0, \frac{1}{4\sqrt{3}}\sqrt{\frac{d}{T}}]$. Notice that this proof holds for $\mathcal{I}_\infty(2\varepsilon)$ class. Scaling down by 2, we have the following for $\mathcal{I}_\infty(\varepsilon)$ class,

$$\mathcal{R}_{\mathcal{I}_\infty(\varepsilon)}(M, T) = \mathcal{R}_{\mathcal{I}_\infty(2 \cdot \frac{\varepsilon}{2})}(M, T) \geq \frac{\varepsilon MT\sqrt{d}}{8}, \text{for } \varepsilon \in \left[ 0, \frac{1}{2\sqrt{3}}\sqrt{\frac{d}{T}} \right].$$

Observe that $\frac{\varepsilon MT\sqrt{d}}{8}$ is a strictly increasing function w.r.t. $\varepsilon$. It is at most $\frac{dM\sqrt{T}}{16\sqrt{3}}$, when $\varepsilon = \frac{1}{2\sqrt{3}}\sqrt{\frac{d}{T}}$. In other words, $\mathcal{R}_{\mathcal{I}_\infty(\varepsilon)}(M, T) \geq \min(\frac{\varepsilon MT\sqrt{d}}{8}, \frac{dM\sqrt{T}}{16\sqrt{3}})$ for $\varepsilon \in [0, \frac{1}{2\sqrt{3}}\sqrt{\frac{d}{T}}]$.

Recall that $\inf_\mathcal{A} \sup_{\mathcal{I} \in \mathcal{I}_\infty(\varepsilon)} \mathcal{R}(M, T) \geq \inf_\mathcal{A} \sup_{\mathcal{I} \in \mathcal{I}_\infty(\varepsilon')} \mathcal{R}(M, T)$, for any $\varepsilon \geq \varepsilon' \geq 0$. Thus, we conclude, $\forall \varepsilon \geq 0$, the following holds,

$$\inf_\mathcal{A} \sup_{\mathcal{I} \in \mathcal{I}_\infty(\varepsilon)} \mathcal{R}_{\mathcal{A}, \mathcal{I}}(M, T) \geq \min(\frac{\varepsilon MT\sqrt{d}}{8}, \frac{dM\sqrt{T}}{16\sqrt{3}}),$$

which completes the proof. $\square$

## C Supporting Lemmas

**Lemma C.1.** *([Lattimore and Csaba, 2020, Theorem 24.2]). Assume $d \leq 2n$ and let $\mathcal{D} = \{x \in \mathbb{R}^d : \|x\|_2 \leq 1\}$. Then there exists a parameter vector $\theta \in \mathbb{R}^d$ with $\|\theta\|_2^2 = d^2/(48n)$ such that $R_n(\mathcal{D}, \theta) \geq d\sqrt{n}/(16\sqrt{3})$.*

**Lemma C.2.** *([Abbasi-Yadkori et al., 2011, Lemma 12]). Let $A, B$ and $C$ be positive semi-definite matrices such that $A = B + C$. Then, we have that*

$$\sup_{x \neq 0} \frac{x^\top A x}{x^\top B x} \leq \frac{\det(A)}{\det(B)}.$$

**Lemma C.3.** *([Abbasi-Yadkori et al., 2011, Lemma 11]). Let $\{X_t\}_{t=1}^{\infty}$ be a sequence in $\mathbb{R}^d$, $V$ a $d \times d$ positive definite matrix and define $\bar{V}_t = V + \sum_{s=1}^{t} X_s X_s^{\top}$. Then, we have that*

$$\log\left(\frac{\det\left(\bar{V}_n\right)}{\det(V)}\right) \leq \sum_{t=1}^{n} \|X_t\|_{\bar{V}_{t-1}^{-1}}^2 .$$

*Further, if $\|X_t\|_2 \leq L$ for all t, then*

$$\sum_{t=1}^{n} \min\left\{1, \|X_t\|_{\bar{V}_{t-1}^{-1}}^2\right\} \leq 2\left(\log \det\left(\bar{V}_n\right) - \log\det V\right) \leq 2\left(d\log\left(\left(\text{trace}(V) + nL^2\right)/d\right) - \log\det V\right)$$

**Lemma C.4.** *(Self-Normalized Bound for Vector-Valued Martingales, [Abbasi-Yadkori et al., 2011, Theorem 1]). Let $\{F_t\}_{t=0}^{\infty}$ be a filtration. Let $\{\eta_t\}_{t=1}^{\infty}$ be a real-valued stochastic process such that $\eta_t$ is $F_t$-measurable and $\eta_t$ is conditionally $R$-sub-Gaussian for some $R \geq 0$ i.e.*

$$\forall \lambda \in \mathbb{R} \quad \mathbf{E}\left[e^{\lambda \eta_t} \mid F_{t-1}\right] \leq \exp\left(\frac{\lambda^2 R^2}{2}\right).$$

*Let $\{X_t\}_{t=1}^{\infty}$ be an $\mathbb{R}^d$-valued stochastic process such that $X_t$ is $F_{t-1}$-measurable. Assume that $V$ is a $d \times d$ positive definite matrix. For any $t \geq 0$, define $\bar{V}_t = V + \sum_{s=1}^{t} X_s X_s^{\top}$. Then, for any $\delta > 0$, with probability at least $1 - \delta$, for all $t \geq 0$,*

$$\left\|\sum_{s=1}^{t} \eta_s X_s\right\|_{\bar{V}_t^{-1}}^2 \leq 2R^2 \log\left(\frac{\det\left(\bar{V}_t\right)^{1/2}\det(V)^{-1/2}}{\delta}\right).$$

**Lemma C.5.** *(Pinsker's inequality [Lattimore and Csaba, 2020, Equation 14.12])*

*For measures $P$ and $Q$ on the same probability space $(\Omega, \mathcal{F})$, we have the following,*

$$\sup_{A \in \mathcal{F}} P(A) - Q(A) \leq \sqrt{\frac{1}{2}D_{KL}(P\|Q)}.$$

