# OpenReview forum: "Multi-Agent Learning with Heterogeneous Linear Contextual Bandits"
_NeurIPS.cc/2023/Conference — NeurIPS 2023 poster_

### Official Review · Reviewer_kifo · 2023-07-06

**Soundness:** 3 good
**Presentation:** 3 good
**Contribution:** 3 good
**Rating:** 6
**Confidence:** 3

**Summary:**

This paper studies the heterogeneous multi-agent contextual bandit problems. They propose the H-LinUCB algorithm, where agents coordinate at the beginning stage by pooling and synchronizing the data, until a certain time determined by the dissimilarity. They show the algorithm is optimal when the tasks are highly similar or highly dissimilar.

**Strengths:**

The paper is well-written and organized in general. Multi-agent learning, especially in the heterogeneous setting, is of great interest in the literature. The scenario considered is hard, as they don't add special structures for the parameters other than assuming the dissimilarity is known. Yet the results are strong and seem to be theoretically rigorous. Moreover, this work improves DISLINUCB by discarding coordination when the dissimilarity level is too high and the coordination hurts the regret.

**Weaknesses:**

The experiment part is relatively weak to me. Currently, the authors consider three settings of different levels of dissimilarity to highlight the advantage of the proposed H-LINUCB in achieving good regret in all settings. However, there isn't a setting where H-LINUCB dominates the other two algorithms. This makes H-LINUCB more likely an interpolation of the fully communicating and fully independent methods. I hope the authors could at least show some experiments where H-LINUCB uniquely succeeds or achieves the best regret, or discuss why this is impossible or hard.

**Questions:**

Although both plots (c) and (d) correspond to regime (ii), H-LINUCB performs differently in the two settings. It resembles Ind-OFUL in (c) but not in (d). What's the key component that leads to the different behavior? Is there a threshold for $\epsilon$ above which H-LINUCB could be shown to copy Ind-OFUL?

**Limitations:**

I don't see any limitations or potential negative societal impact of this work.

---

> ### Author Rebuttal · Authors · 2023-08-03
>
> We thank you for your feedback. We address your concern about the experiments as follows:
>
> > The experiment part is relatively weak to me. Currently, the authors consider three settings of different levels of dissimilarity to highlight the advantage of the proposed H-LINUCB in achieving good regret in all settings. However, there isn't a setting where H-LINUCB dominates the other two algorithms. This makes H-LINUCB more likely an interpolation of the fully communicating and fully independent methods. I hope the authors could at least show some experiments where H-LINUCB uniquely succeeds or achieves the best regret, or discuss why this is impossible or hard.
>
> We believe that the experiments efficiently demonstrate our objectives. Our goal was to show that H-LinUCB is essentially behaves as “the best of the both worlds” approach. Specifically, our experiment aims to show that H-LinUCB can improve regret when there are opportunities for collaboration, and does not collaborate if it gains no benefit. Therefore, it behaves as DisLinUCB when it is beneficial to collaborate and behaves as Ind-OFUL, otherwise.
>
> On the contrary, Ind-OFUL is unable to leverage similarity structure between bandits to learn faster, whereas DisLinUCB incurs linear regret in the case that collaboration only hurts the performance. Thus, DisLinUCB and Ind-OFUL both fail in different ways.
>
> Regarding your question on the interpolation, for small $\epsilon$, we would see that H-LinUCB performs worse than DisLinUCB due to the artifact of the extra term $\alpha \epsilon Mt$ to handle the dissimilarity (Lemma 4.1).
>
> > Although both plots (c) and (d) correspond to regime (ii), H-LINUCB performs differently in the two settings. It resembles Ind-OFUL in (c) but not in (d). What's the key component that leads to the different behavior?
>
> The reason is that $\epsilon$ increases significantly from plots (c) to (d). In plot (d), H-LinUCB only collaborates for a small number of rounds, and switches to independent learning; the plot therefore looks nearly identical to Ind-OFUL.
>
> > Is there a threshold for $\epsilon$ above which H-LINUCB could be shown to copy Ind-OFUL?
>
> From a theoretical viewpoint, for $\epsilon=1$, H-linUCB will behave like Ind_OFUL. However, experimentally, it is not clear what that threshold is – it would vary from experiment to experiment due to a randomly generated set of parameters. For some large enough $\epsilon$ (could be smaller than 1, e.g., plot (d) ), H-LinUCB only allows agents to collaborate for a small number of rounds then switches to learn independently, so the regret of H-LinUCB is nearly identical to that of Ind-OFUL by just looking at the plot.

---

> > ### Comment · Reviewer_kifo · 2023-08-12
> >
> > Thank you for addressing my concerns! I'll keep my original score.

---

### Official Review · Reviewer_gcmR · 2023-07-06

**Soundness:** 3 good
**Presentation:** 2 fair
**Contribution:** 2 fair
**Rating:** 5
**Confidence:** 3

**Summary:**

This paper considers a multi-agent linear contextual bandit (with central server coordination) setting where $M$ agents each has (possibly different) unknown but fixed $d$-dimensional bandit parameter $\theta_m$, and the $\ell_2$-norm bound on the bandit parameters $\epsilon$ (i.e., $||\theta_i - \theta_j||_2 \leq \epsilon, \forall i, j \in [M]$) is known to the agents. The authors propose a UCB-based algorithm that collaborates for the first $\epsilon^{-2}$ time slots and then learns individually. The authors also present the related regret upper bound and lower bound and run experiments on synthetic data.

**Strengths:**

- Heterogeneous collaborative multi-agent setting is an upcoming area of interest, so this paper is likely to be relevant to the community.
- The idea of adjusting the length of collaboration by the dissimilarity (i.e., $\tau = \min(\lfloor\epsilon^{-2}\rfloor, T)$) is neat and intuitive.
- The code used for the experiments is also included and the results should be easily replicable.

**Weaknesses:**

- The introduction of the proposed algorithm (Section 4.1) is not easy to follow and understand, many new variables are used without explanations
- Notion of heterogeneity and the part of algorithm for handling collaboration with dissimilarity seem to be straightforward extensions of prior works, the new technical challenges for analyzing the proposed model and algorithm are not clearly discussed

**Questions:**

- How does the algorithm perform if the given $\epsilon$ is off from the real $\epsilon$ of the environment?
- How does the result in this paper generalize to linear (non-contextual) bandit?
- How important is the dissimilarity defined on $\ell_2$-norm? Why didn't the authors consider $\ell_\infty$ norm as prior work?
- (minor point) It is not clear what kind of norm the one in line 42 is.


**Limitations:**

The authors discuss some limitations (that $\epsilon$ is assumed known and that the proposed algorithm is not optimal for some values of $\epsilon$) in the conclusion section. In my opinion, the primary limitation of this work is the practicality of the proposed algorithm as $\epsilon$ is often not precisely known in real application.

---

> ### Author Rebuttal · Authors · 2023-08-03
>
> We thank you for your suggestion. We address your concerns as follow:
>
> > The introduction of the proposed algorithm (Section 4.1) is not easy to follow and understand, many new variables are used without explanations
>
> In line 196, we define these variables as sufficient statistics. Specifically, $V$ is the gram matrix, and $b$ is the sum of the product of reward and context vector, which we use to form a regularized least squares solution and construct the confidence bound for our linear bandit problem.
>
> > How does the algorithm perform if the given $\epsilon$ is off from the real $\epsilon$ of the environment?
>
> As we discussed in our submission, the current scope of our paper is when $\epsilon$ is known and we leave the case when $\epsilon$ is unknown to a future work.
>
> > How does the result in this paper generalize to linear (non-contextual) bandit?
>
> Our setting is linear bandit. The arms in the decision set are also referred to as the contexts in the literature.
>
> > How important is the dissimilarity defined on $\ell_2$-norm? Why didn't the authors consider $\ell_\infty$-norm as prior work?
>
> It is not important. There is no particular reason why we choose $\ell_2$ over $\ell_\infty$, it is just natural to use $\ell_2$-norm to define the distance of two vectors. When using $\ell_\infty$-norm the results would change by a factor of $\sqrt{d}$. Even if we use $\ell_\infty$-norm, it is also not comparable to the prior work of Wang et al. 2021,  because (1) $\epsilon$ hides inside the subpar-arm set, and (2) it is not possible for us to define such subpar arm set in our setting when the arm sets can be generated arbitrarily.
>
> > (minor point) It is not clear what kind of norm the one in line 42 is.
>
> It is $\ell_2$-norm. We'll add the subscript in the revision.

---

> > ### Comment · Reviewer_gcmR · 2023-08-14
> >
> > I appreciate the authors' response to my questions and concerns.
> >
> > I would encourage the authors to polish Section 4.1 and make the definitions of variables clearer.
> >
> > I remain my recommendation for this paper.

---

> > > ### Author Response · Authors · 2023-08-20
> > >
> > > We thank you for the suggestions. We will ensure that the definitions of variables are clearer in the next revision.

---

### Official Review · Reviewer_bsff · 2023-07-10

**Soundness:** 3 good
**Presentation:** 3 good
**Contribution:** 2 fair
**Rating:** 6
**Confidence:** 4

**Summary:**

This paper studies multi-agent linear stochastic bandit under a model of heterogeneity of the agents. Concretely, the model assumes a centralized controller who can communicate information to and from all of the M different agents in the network. At each time, every agent in the network, will play an arm from a set of possible arms (that are adversarially chosen). The choice of which arm an agent plays is a function of only the past information at that agent -- namely the agent's past arms pulled and rewards observed and the information communicated by the central controller to the agent. Subsequently, the agent observes a noisy stochastic reward where the noise is independent across agents and time.

To facilitate group learning, the paper assumes structure among the heterogeneity. Concretely, the paper assumes that the unknown parameter for any pair of agents is at-most \epsilon away in L2 distance. This modeling assumes that although the agents are different, nonetheless have a latent structure which can enable faster learning.

Under this structure, the paper proposes an algorithm such that the sum regret of all agents scale sub-linearly in M and T. The paper also gives lower bound evidencing that their upper bounds are order wise tight.

**Strengths:**

Well articulated problem, algorithm, intuition and the key steps of the analysis.

**Weaknesses:**

The main weakness is the positioning and contributions of the paper in comparison to "Multi-agent heterogeneous stochastic linear bandits, Ghosh et.al. in ECML-PKDD 2022" which the submission cites.

Specifically, the submission claims that the model of heterogeneity studied is different from Ghosh et.al. (Line 280 of the submission). However, a cursory examination indicates that the assumption of the submission is stronger than Ghosh et.al's in the sense that Defn 3.1 of the submission implies that the average parameter is close to any agent's parameter by \epsilon, which is the assumption in Ghosh et.al. Does this imply that the assumptions in the present paper with regard to heterogeneity are stronger than Ghosh et.al.'s?

A somewhat less satisfying feature of the algorithm is also the known \epsilon which seems to be non needed by Ghosh et.al. ?

**Questions:**

The main questions for the authors is to elucidate their contributions in comparision to Ghosh et.al.?

Specifically,

1)The authors make a stronger assumption on the heterogeneity (Defn 3.1) compared to Ghosh et.al. Does this stronger assumption on heterogeneity is what enables the authors to allow for adversarially generated set of arms?

2) The algorithm of Ghosh et.al. does not need \epsilon while the present paper assumes knowledge of \epsilon. Can the authors comment on the necessity of this information?

3) In the case of stochastic arms, is the  weaker result of the submission compared to Ghosh et.al. the price needed to pay for having an adversarial arm setup?

***Post Rebuttal: *** Thank you for your response and I have updated my score.

---

> ### Author Rebuttal · Authors · 2023-08-03
>
> We thank you for your insightful feedback and questions. We hope to clarify our contributions compared to Ghosh et al. as follows:
>
> Besides the stochastic assumption, Ghosh et al. also make a **strong distributional assumption** on how the arm is being generated. Essentially, they require a lower bound on the smallest eigenvalue of the covariance matrix of context vectors. Formally, they need that  $E_{t-1}[\beta_{i,t}\beta_{i,t}^\top] \geq \rho_{\min} I$ ; see Equation~1 of Ghosh et al.
> This assumption is crucial for them as it allows them to estimate the average parameter and motivates the algorithm design.
>
> We do not make any such distributional assumption since it is often unrealistic in practical settings. On the contrary, we consider an adversarial setting where the arm sets at each round can be selected by an (oblivious) adversary and the size of the arm sets can even be infinite. Our setting renders the algorithmic design and guarantees of Ghosh et al. inapplicable as far as we can tell, thus requiring a completely new treatment.
>
> > Specifically, the submission claims that the model of heterogeneity studied is different from Ghosh et.al. (Line 280 of the submission). However, a cursory examination indicates that the assumption of the submission is stronger than Ghosh et.al's in the sense that Defn 3.1 of the submission implies that the average parameter is close to any agent's parameter by \epsilon, which is the assumption in Ghosh et.al. Does this imply that the assumptions in the present paper with regard to heterogeneity are stronger than Ghosh et.al.'s?
>
> Yes, our notion of heterogeneity is slightly stronger.  However, we can possibly work with the same heterogeneity assumption as in Ghosh et al. This is simply a matter of presentation. In particular, note that for their personalized framework, Ghosh et al. give a regret bound for individual agents – it is, therefore, natural to have a more fine-grained notion of heterogeneity by defining the dissimilarity of the individual parameter with the average parameters. However, our work emphasizes the collaboration among the agents, and we are interested in the _group_ regret instead of individual regret. Therefore, the heterogeneous notion which we proposed is a natural way to capture the heterogeneity among the agents. Again, these differences are merely cosmetic. We can work with the same assumption as Ghosh et al. and present bounds on the individual agent’s regret.
>
>
> > The authors make a stronger assumption on the heterogeneity (Defn 3.1) compared to Ghosh et.al. Does this stronger assumption on heterogeneity is what enables the authors to allow for adversarially generated set of arms?
>
> No, as we say in the remarks above, the difference between the two notions of heterogeneity is not significant. We do not rely on the heterogeneity assumption to allow for an adversarially generated set of arms.
>
> > The algorithm of Ghosh et.al. does not need \epsilon while the present paper assumes knowledge of \epsilon. Can the authors comment on the necessity of this information?
>
> The knowledge of (an upper bound on) $\epsilon$ facilitates our algorithmic design in constructing the confidence bound and controlling the collaboration between tasks. Adapting to the unknown $\epsilon$ in an adversarial setting that we consider here would require additional work. It is important to note that the adaptation strategy of Ghosh et al. (to unknown $\epsilon$) _fails_ in an adversarial setting.
>
> > In the case of stochastic arms, is the weaker result of the submission compared to Ghosh et.al. the price needed to pay for having an adversarial arm setup?
>
> Yes. However, we would like to reiterate that the two settings are simply _incomparable_. Again, Ghosh et al. need a strong distributional assumption to obtain their bounds. In fact, the $O(T^{1 /4})$ bound that they achieve (for $\epsilon = 0$) is not even possible information-theoretically in an adversarial setting. The minimax lower bound in such settings is $O(\sqrt{T})$ which we match.

---

> > ### Comment · Reviewer_bsff · 2023-08-11
> > **Thank you for the response. Increasing my score**
> >
> > Thank you for this comparison. I would encourage the authors to update Remark 4.1 of the draft with the response provided here -- namely that the present paper improves on restrictive stochastic arm assumption, but pays the price for doing so by needing \epsilon and a worse off regret in the stochastic case compared to Ghosh et.al. However, unlike Ghosh et.al., the present paper yields sub-linear regret bounds in M and T even under adversarial generated arms.
> >
> > PS: I also increased my score from 4 --> 6.

---

> > > ### Author Response · Authors · 2023-08-14
> > >
> > > We thank you for the valuable feedback and re-evaluating the score. We will integrate the discussion in our next revision.

---

### Official Review · Reviewer_5b9x · 2023-07-20

**Soundness:** 3 good
**Presentation:** 2 fair
**Contribution:** 3 good
**Rating:** 6
**Confidence:** 4

**Summary:**

This work considers a multi-agent linear contextual bandit model with heterogeneity among the agents. The authors propose a novel algorithm called H-LinUCB to minimize the group cumulative regret when agents communicate through a central server. When the level of heterogeneity is known to the agents, they show that H-LinUCB is provably optimal in regimes when the agents are highly similar or dissimilar.

**Strengths:**

Originality:
------------
- The authors introduced the notion of heterogeneity as a $\epsilon$-MALCB problem, inspired by (Wang et al. (2021))
- The model considered in this paper is well-motivated
- Related work covered in detail

Quality:
-----------
- The theoretical analysis seems to be concrete
- Numerical simulations are provided to validate the theoretical results

Clarity:
-----------
- The paper is well-written and easy to follow, with the exception of some proofs in the supplementary material

Significance:
-----------
- The proposed notion of heterogeneity could be of interest to researchers, particularly for the case when the level of heterogeneity is unknown

**Weaknesses:**

- The algorithmic ideas and the regret analysis are known extensions of the work in (He et al. (2022), Wang et al. (2020), Dubey et al. (2020))
- The lower bound proof is extended from the ideas in (Lattimore and Csaba (2020))
- The algorithm depends on the assumption that the level of heterogeneity $\epsilon$ is known, which need not be true in realistic scenarios
- The proof of Theorem 4.1 in Appendix A.2 is not easy to follow, because the steps aren't explained in detail. Even though the regret analysis is extended from (He et al. (2022), Wang et al. (2020), Dubey et al. (2020)), I believe a paper should be self-contained.
- The lower bound seems to be loose, because it is off from the regret upper bound by a factor of $d$ in one of the terms. Furthermore, it is not applicable in the regime $\epsilon \in (1/\sqrt{MT}, d/\sqrt{T})$


**Questions:**

- Some of the notation isn't defined in Appendix A.2. What is $V_p$ in line 498? How is the condition on the ratio of determinants obtained below line 499? I was hoping if the authors could re-write the proof of Theorem 4.1 so that it is easy to follow

----

Post rebuttal: I have raised the score from 5 to 6.

**Limitations:**

I don't see any potential negative societal impact of their work

---

> ### Author Rebuttal · Authors · 2023-08-03
>
> We thank you for your time and effort to review our submission. We hope to address your concerns as follows.
>
> > The proof of Theorem 4.1 in Appendix A.2 is not easy to follow, because the steps aren't explained in detail. Even though the regret analysis is extended from (He et al. (2022), Wang et al. (2020), Dubey et al. (2020)), I believe a paper should be self-contained.
>
> > Some of the notation isn't defined in Appendix A.2. What is $V_p$ in line 498?
>
> On line 196, we denote matrix $V_{sync}$ as a sufficient statistic needed to form a least square estimate. Essentially, the matrix $V_{sync}$ contains all the samples after the synchronization (line 20 algorithm 1). In line 495, we have $P$ as total epochs where the synchronization happens. Here, $V_P$ is a matrix that contains all the samples of the agents at the last synchronization epoch. We apologize for not clearly introducing this bit of notation; we will fix that in the next revision.
>
> > How is the condition on the ratio of determinants obtained below line 499? I was hoping if the authors could re-write the proof of Theorem 4.1 so that it is easy to follow
>
> Sure, we will be happy to provide those details. Essentially,
> by telescoping, we have that $\log \frac{\det(V_P)}{\det(V_0)}=\log \frac{\det(V_P) \cdot \det(V_{P-1}) \cdots \det(V_{1})}{\det(V_{P-1}) \cdots \det(V_{1}) \cdot \det(V_0)}=\sum_{p=1}^P\log \frac{\det(V_p)}{\det (V_{p-1})}$.
> Furthermore, we have the following condition (which we state above line 499):
>
> $\log \frac{\det(V_P)}{\det(V_0)}\leq d \log ( 1+ \frac{M\tau }{\lambda  d } )$
>
> Therefore, we have at most $R = \lceil d \log ( 1+ \frac{M\tau }{\lambda  d } )\rceil$ epochs such that $\log \frac{\det(V_p)}{\det(V_{p-1})}\geq 1$; otherwise, we have $\log \frac{\det(V_P)}{\det(V_0)}> d \log ( 1+ \frac{M\tau }{\lambda  d } )$, which violates the condition above. This implies that for all but $R$ epochs, we have that
> $0 \leq \log\frac{\det(V_p)}{\det(V_{p-1})}\leq 1$, which is $1 \leq \frac{\det(V_p)}{\det(V_{p-1})}\leq 2$.
>
> We thank you for your suggestion. We will add the explanations to the proof of Theorem 4.1. in our revision.

---

> > ### Comment · Reviewer_5b9x · 2023-08-11
> > **Response to the rebuttal**
> >
> > I thank the authors for answering my questions. A minor comment: it seems that the $\log$ in this submission is the natural logarithm, in that case, shouldn't $\frac{\det(V_p)}{\det(V_{p-1})} \leq e$?
> >
> > I raise my score from 5 to 6, and it would be greatly appreciated if the authors add the explanations to the proof of Theorem 4.1 in their revision.

---

> > > ### Author Response · Authors · 2023-08-14
> > >
> > > We thank you for going through the discussion and raising the score. Your suggestion would indeed make the proof easier to follow.
> > >
> > > Regarding your question on the ratio, we use logarithm base 2 for that determinant ratio. Using natural logarithm would only change the bound by a constant factor, we will add this clarification in the revision.

---

### Official Review · Reviewer_JJk8 · 2023-07-26

**Soundness:** 3 good
**Presentation:** 3 good
**Contribution:** 3 good
**Rating:** 6
**Confidence:** 3

**Summary:**

This paper studies multi-agent linear contextual bandits problem where each agent faces different bandits model. The heterogeneity is captured by the $l_2$ distance between the environment parameters. An upper confidence bound (UCB) algorithm, termed as H-LinUCB is proposed. The regret upper bound nearly-matches the lower bound proved by this paper in both small and large heterogeneity regime. The experimental results also valid the effectiveness of the proposed algorithm.

**Strengths:**

+ The paper provides solid theoretical results including both regret upper bound and regret lower bound. The discussion of the small and large $\epsilon$ cases are insightful.

+ The paper provides experimental results to validate their theoretical results.


**Weaknesses:**

The major weaknesses is the design of the stopping criterion. Please see the question part for detailed discussion.

**Questions:**

It seems like we can have a simpler criterion when $\epsilon$ is known. For example, if $\epsilon\leq 1/\sqrt{T}$, we deploy a H-LinUCB with $\tau=T$ and when $\epsilon\geq 1/\sqrt{T}$, we deploy individual OFUL algorithm for each agent. The regret upper bound is $O(\epsilon MT + d\sqrt{MT})$ and $dM\sqrt{T}$ respectively, which scales the same as that provided in the paper. In my opinion, this rule is simple and may have lower communication and computation cost since there is no communication when $\epsilon$ is large.

Can author discuss whether this method works?

Another question is that is it able to detect the magnitude of $\epsilon$ when $\epsilon$ is unknown so that the algorithm can automatically “stop” to aggregate?

---

> ### Author Rebuttal · Authors · 2023-08-03
>
> We thank you for your suggestions. We address your concerns and questions as follows:
>
> >.... Can author discuss whether this method works?
>
> We appreciate your discussion and thank you for raising an insightful question. Mathematically, in the scenario where $\epsilon$ is known, your proposed rule would indeed yield the same statistical guarantees. However, our objective was to design a generic algorithm that would be adaptive to an unknown $\epsilon$, even though in our current results, we only worked in the known $\epsilon$ setting. Specifically, our stopping rule adjusts the collaboration based on the $\epsilon$, enabling us to extend the current algorithm to the case of unknown $\epsilon$.
>
> > Another question is that is it able to detect the magnitude of when $\epsilon$  is unknown so that the algorithm can automatically “stop” to aggregate?
>
> That is also an excellent question, something we leave for future work. We do believe that it is possible to estimate $\epsilon$ (by doubling trick, corralling, etc.) using data collected during early rounds. Depending on the estimate, the algorithm would determine whether it will continue to collaborate or cease it. We note, though, that what we really require is any upper bound on $\epsilon.$

---

> > ### Comment · Reviewer_JJk8 · 2023-08-12
> > **Thank you for the rebuttal**
> >
> > Thank you for answering my previous questions. I decide to keep my original score.

---

### Decision · Program_Chairs · 2023-09-21

**Decision:**

Accept (poster)

**Comment:**

Reviewers all agreed that the problem studied is interesting, and the technical results are solid. There were some concerns regarding the contributions of this paper in comparison to Ghosh et.al. and several other previous works. The authors' rebuttal convinced the reviewer about the novelty of this paper. After the rebuttal, all reviewers view this submission positively.